# Inositol polyphosphate multikinase physically binds to the SWI/SNF complex and modulates BRG1 occupancy in mouse embryonic stem cells

**Jiyoon Beon[1†], Sungwook Han[1†], Hyeokjun Yang[1], Seung Eun Park[1], Kwangbeom Hyun[1], Song-Yi Lee[2], Hyun-Woo Rhee[2], Jeong Kon Seo[3], Jaehoon Kim[1], Seyun Kim[1,4,5]*, Daeyoup Lee[1]***

[1]Department of Biological Sciences, Korea Advanced Institute of Science and Technology (KAIST), Daejeon, Republic of Korea; [2]Department of Chemistry, Seoul National University, Seoul, Republic of Korea; [3]UNIST Central Research Facilities (UCRF), Ulsan National Institute of Science and Technology (UNIST), Ulsan, Republic of Korea; [4]KAIST Stem Cell Center, KAIST, Daejeon, Republic of Korea; [5]KAIST Institute for the BioCentury, KAIST, Daejeon, Republic of Korea

*For correspondence:
seyunkim@kaist.ac.kr (SK);
daeyoup@kaist.ac.kr (DL)

†These authors contributed equally to this work

**Abstract** Inositol polyphosphate multikinase (IPMK), a key enzyme in inositol polyphosphate (IP) metabolism, is a pleiotropic signaling factor involved in major biological events, including transcriptional control. In the yeast, IPMK and its IP products promote the activity of the chromatin remodeling complex SWI/SNF, which plays a critical role in gene expression by regulating chromatin accessibility. However, the direct link between IPMK and chromatin remodelers remains unclear, raising the question of how IPMK contributes to transcriptional regulation in mammals. By employing unbiased screening approaches and in vivo/in vitro immunoprecipitation, here we demonstrate that mammalian IPMK physically interacts with the SWI/SNF complex by directly binding to SMARCB1, BRG1, and SMARCC1. Furthermore, we identified the specific domains required for IPMK-SMARCB1 binding. Notably, using CUT&RUN and ATAC-seq assays, we discovered that IPMK co-localizes with BRG1 and regulates BRG1 localization as well as BRG1-mediated chromatin accessibility in a genome-wide manner in mouse embryonic stem cells. Together, these findings show that IPMK regulates the promoter targeting of the SWI/SNF complex, thereby contributing to SWI/SNF-meditated chromatin accessibility, transcription, and differentiation in mouse embryonic stem cells.

## Editor's evaluation

This study describes a physical interaction between the Inositol polyphosphate multikinase enzyme (IPMK) and the SWI/SNF chromatin-remodeling complex. IPMK modulates SWI/SNF chromatin binding in mouse embryonic stem cells in particular at the transcription start sites of promoters with bivalent chromatin modifications. Loss of IPMK function led to defective activation of endodermal genes upon cell differentiation. This study identifies IPMK as a novel regulator of chromatin remodeling by SWI/SNF and gene expression in embryonic stem cells.

## Introduction

Inositol polyphosphates (IPs) are a class of signaling messengers that mediate diverse biological events, such as cellular growth, proliferation, and metabolic homeostasis. Inositol polyphosphate multikinase (IPMK) is an essential enzyme for the synthesis of IPs, including inositol tetrakisphosphates

[IP$_4$, both Ins(1,3,4,5)P$_4$ and Ins(1,4,5,6)P$_4$] and pentakisphosphates [IP$_5$, Ins(1,3,4,5,6)P$_5$] (*Chakraborty et al., 2011*; *Hatch and York, 2010*; *Saiardi et al., 1999*). In addition to its role as a phosphatidylinositol 3-kinase [thereby producing phosphatidylinositol 3,4,5-trisphosphate (PIP$_3$)], IPMK also non-catalytically controls the activity of various signaling factors, including mTOR, AMPK, and TRAF6 (*Bang et al., 2012*; *Kim et al., 2017a*; *Kim et al., 2017b*; *Kim et al., 2011*; *Maag et al., 2011*; *Resnick et al., 2005*). These findings suggest that IPMK plays a critical role in coordinating major biological events.

Increasing evidence strongly indicates that nuclear IPMK acts as a key factor in the regulation of gene expression. The gene encoding IPMK was originally cloned from yeast as a gene required for the regulation of arginine metabolism and named *Arg82* (yeast *Ipmk*) (*Bechet et al., 1970*; *Dubois et al., 1987*; *Odom et al., 2000*). The physical interaction between Arg82 and yeast transcription factor MCM1, a yeast homolog of mammalian SRF, is crucial for transcriptional control in yeast (*Bercy et al., 1987*; *Christ and Tye, 1991*; *Messenguy and Dubois, 1993*; *Odom et al., 2000*). In mammals, IPMK-SRF binding is a critical event for the SRF-dependent induction of gene expression (*Kim et al., 2013*). Other functions of nuclear IPMK are mediated by its diverse interactions with, for example, p53, steroidogenic factor 1, and CBP/p300 (*Blind, 2014*; *Blind et al., 2014*; *Malabanan and Blind, 2016*; *Xu et al., 2013a*; *Xu et al., 2013c*; *Xu and Snyder, 2013b*).

Chromatin remodeling is essential for efficient transcription of eukaryotic genes (*Kouzarides, 2007*; *Trotter and Archer, 2007*; *Vignali et al., 2000*). In particular, SWI/SNF is a large family of ATP-dependent chromatin remodeling complexes characterized as transcriptional regulators. These complexes enable the transcription machinery or other transcription factors to access target genes (*Arnaud et al., 2018*; *Hargreaves and Crabtree, 2011*). In mammalian cells, the canonical SWI/SNF complex contains one of the two mutually exclusive ATPases, BRM (SMARCA2) or BRG1 (SMARCA4), in addition to a core set of subunits consisting of BAF155 (BRG1-associated factor or SMARCC1), SMARCB1 (hSNF5 or INI1), and BAF170 (SMARCC2), as well as four to eight accessory subunits (*Khavari et al., 1993*; *Wang et al., 1996a*; *Wang et al., 1996b*). Importantly, the SWI/SNF complex mediates modifications on the nucleosome structure and regulates nucleosome positioning in an ATP-dependent manner, thereby modulating the accessibility of regulatory proteins. Therefore, this chromatin remodeling complex is critical for various biological processes, including gene transcription, cell-cycle regulation, and cell differentiation (*Ho et al., 2009*; *Ho et al., 2009*; *Hodges et al., 2016*; *Kim and Roberts, 2014*; *Tolstorukov et al., 2013*; *Wang et al., 2014*).

Despite the importance of both IP function and chromatin remodeling in transcriptional regulation, only a few studies have addressed the link between IPs and chromatin remodeling. A previous study in yeast demonstrated that IPs can regulate the nucleosome-sliding activity of chromatin remodeling complexes in vitro (*Shen et al., 2003*). Specifically, IP$_4$ and IP$_5$ stimulate the activity of the SWI/SNF complex, whereas inositol hexakisphosphate (IP$_6$) inhibits the activity of the NURF, ISW2, and INO80 complexes. Another study in yeast illustrated that mutation in the yeast IPMK homolog Arg82 impairs IP$_4$ and IP$_5$ production and causes inefficient recruitment of the SWI/SNF complex and impaired chromatin remodeling at the promoter of the phosphate-responsive gene *PHO5* (*Steger et al., 2003*). In mammals, inositol hexakisphosphate kinase 1 (IP6K1) directly interacts with histone demethylase JMJD2C. IP6K1 and its product 5-IP$_7$ appear to mediate the expression of JMJDC2 target genes in mammalian cells by regulating the chromatin association of JMJDC2 and the levels of trimethyl-histone H3 lysine 9 (*Burton et al., 2013*). Taken together, these findings suggest that IPs and relevant enzymes play important roles in chromatin remodeling and transcription. However, a direct link between IPMK (which produces IPs) and the chromatin remodeling complex SWI/SNF has not been reported, and whether IPMK contributes to transcriptional regulation in mammals is unclear.

To address these points, we performed unbiased screening assays and elucidated that the core subunits of the SWI/SNF complex, including SMARCB1 and BRG1, physically interact with IPMK. The physical association between IPMK and the SWI/SNF complex was confirmed via in vitro and in vivo immunoprecipitation assays. The specific binding sites between IPMK and SMARCB1 were also mapped in detail. To investigate the biological role of the interaction between IPMK and the SWI/SNF complex, we performed next-generation sequencing. We also examined the co-localization of IPMK and BRG1, finding that they co-localize at the chromatin, especially at the promoter-transcriptional start site (TSS) and enhancers. Surprisingly, IPMK depletion significantly reduces the genomic occupancy of BRG1 and BRG1-mediated chromatin accessibility, especially at bivalent promoters. IPMK

depletion also affects gene transcription, with reduced BRG1 occupancy and chromatin accessibility at the promoter-TSS. Finally, we demonstrate that IPMK depletion significantly impacts endodermal gene expression during differentiation. Taken together, our findings indicate direct linkage between IPMK and the SWI/SNF complex via physical interactions, as well as a crucial role of IPMK in regulating the genomic occupancy of BRG1 and BRG1-associated chromatin accessibility, transcription, and stem cell differentiation.

## Results

### Identification of IPMK-binding/interacting proteins

To identify IPMK targets, we performed yeast two-hybrid screening with IPMK as the bait and a human brain cDNA library as the prey. The co-transformants of GAL4-DB fusion plasmid pGBKT7-IPMK (prey) and GAL4-AD fusion plasmid pACT2-SMARCB1 (bait) activated expression of the reporter gene, demonstrating cell growth on selective media, unlike the co-transformants of pGBKT-7 and pACT2-SMARCB1 (*Figure 1A*). Approximately 23–36 proteins were identified as potential interactors with IPMK (*Supplementary file 1*). Among these putative targets, only SMARCB1 was present in the results for both duplicates of the yeast two-hybrid screening (*Supplementary file 1*). These results indicate SMARCB1, a core subunit of the SWI/SNF chromatin remodeler, as a potential IPMK-binding protein.

To further identify potential target proteins that interact with IPMK in mammalian cells, we employed an in vivo proximity-labeling approach using an engineered variant of ascorbate peroxidase (APEX2) fused to IPMK (APEX2-mediated proximity labeling) in human embryonic kidney (HEK)–293 cells. The proteins in the vicinity of the IPMK-APEX2 fusion protein or those of APEX2 alone were biotinylated, enriched using streptavidin beads, and identified via mass spectrometry (*Figure 1B*). A total of 455 IPMK-associated candidate proteins were identified by comparing the proteins in the vicinity of APEX2-IPMK to those of APEX2 alone (background, used as a negative control) based on twofold differences in the enrichment score. Interestingly, ConsensusPathDB (*Herwig et al., 2016*) using the IPMK-associated candidates identified the enriched protein complex-based sets related to BRG1-, BAF-, or SWI/SNF complex-associated complexes (*Supplementary file 2*). Among these candidates, we detected factors associated with the SWI/SNF complex, including SMARCB1 (BAF47), BRG1 (SMARCA4), SMARCC2 (BAF170), ARID1A (BAF250A), PBRM1 (BAF180), and SMARCC1 (BAF155), as IPMK interactors (*Figure 1C*). Furthermore, among these SWI/SNF complex subunits, SMARCB1 had the most significant interaction with IPMK or was positioned very proximal to IPMK (*Figure 1C*), consistent with the yeast two-hybrid screening results (*Figure 1A*). We also detected core histones (histones H2B, H3.1, and H4) as IPMK interactors (*Figure 1—figure supplement 1*), supporting that IPMK interacts with the SWI/SNF complex, which binds to nucleosomes in vivo. Collectively, these results from two unbiased screening experiments strongly indicate that IPMK physically interacts with the SWI/SNF complex.

### Physical interaction between IPMK and the core subunits of the SWI/SNF complex

To validate the physical interaction between IPMK and SMARCB1, we performed an in vitro binding assay using recombinant IPMK and SMARCB1 proteins. We detected a direct protein-protein interaction between IPMK and SMARCB1 (*Figure 2A*, see lane 2 and lane 4). To confirm the physical association between IPMK and the core subunits of the SWI/SNF (BAF) complex, we performed binary protein-interaction assays with baculovirus-mediated expression. We co-infected Sf9 insect cells with baculoviruses expressing FLAG-IPMK and individual subunits of the SWI/SNF complex, including SMARCB1, BRG1, BAF155 (SMARCC1), and BAF170 (SMARCC2). Next, we performed FLAG M2 agarose immunoprecipitation followed by immunoblotting. We detected direct interaction of IPMK with each of the SMARCB1, BRG1, and BAF155 proteins, but not with BAF170 (*Figure 2B*, see lane 7 and lane 8). Taken together, these results indicate that IPMK can bind to SMARCB1, BRG1, or BAF155 in vitro.

To investigate whether the interactions we observed also occur in vivo, we performed co-immunoprecipitation experiments with mammalian cells. We detected a direct association between endogenous IPMK and SMARCB1 in mouse embryonic stem cells (mESCs; *Figure 2C and D*) and in mouse embryonic fibroblasts (MEFs; *Figure 2—figure supplement 1A-C*). Consistent with our in

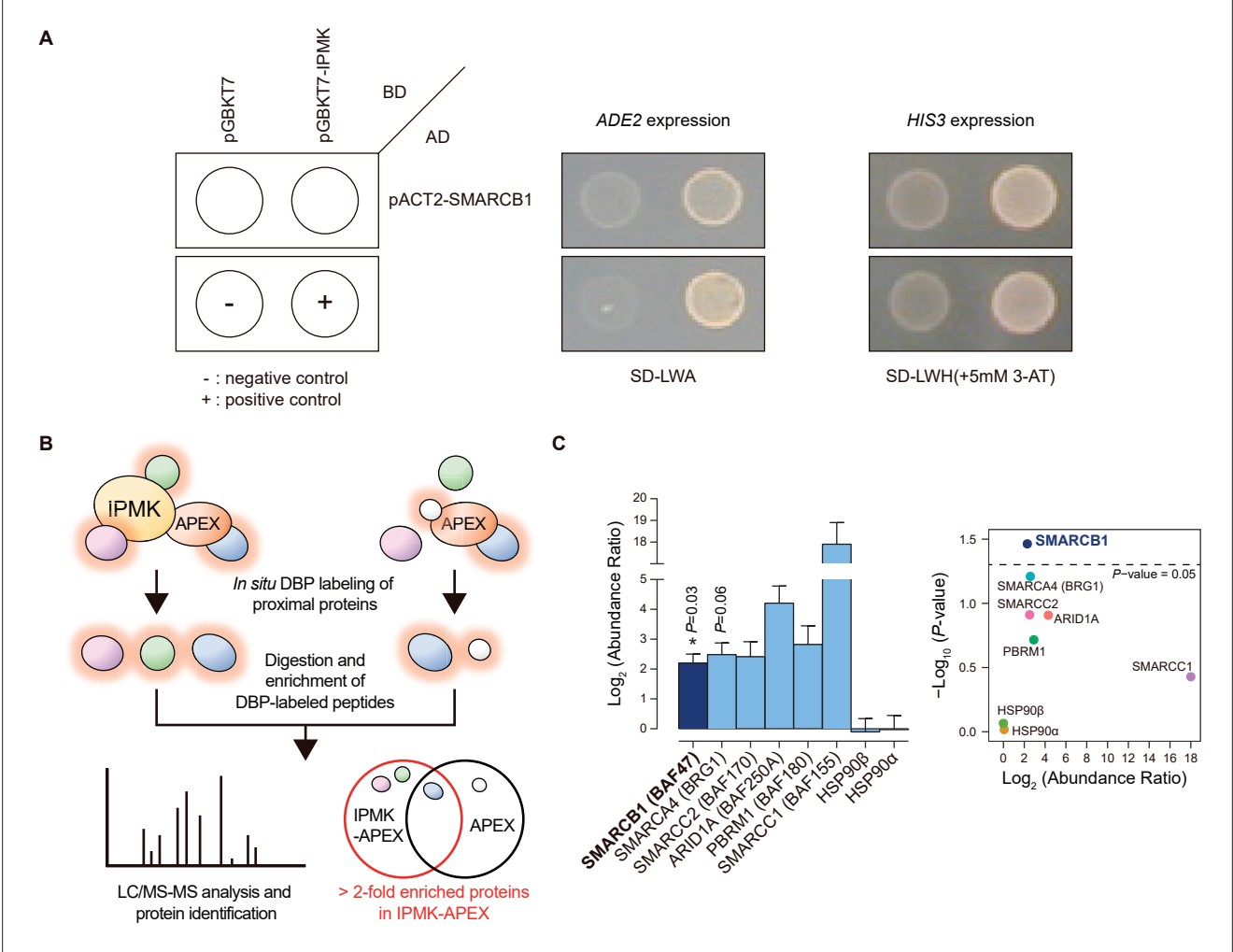

**Figure 1.** Identification of SMARCB1 as an IPMK-interacting protein via unbiased screening assays. (**A**) IPMK and SMARCB1 interaction test in yeast strain AH109 containing two reporter genes (*ADE2* and *HIS3*). Yeast cells were co-transformed with either the GAL4-BD fusion plasmid pGBKT7 or pGBKT7-IPMK, and the GAL4-AD fusion plasmid pACT2-SMARCB1. The yeast cells were spread on a selection medium lacking leucine and tryptophan (SD-LW) to select co-transformants of bait and prey vectors. Specific interactions between bait and prey proteins were monitored via cell growth on selection medium lacking leucine, tryptophan, and adenine (SD-LWA), or a selection medium lacking leucine, tryptophan, and histidine (SD-LWH). 3-Amino-1,2,4-triazole (3-AT) was used to suppress leaky *HIS3* expression in transformants to obtain an accurate phenotype. Polypyrimidine tract binding protein (PTB) gene fused with the GAL4 DNA binding domain (BD-PTB) and PTB gene fused with the GAL4 activation domain (AD-PTB) were used as positive controls of bait and prey vectors, respectively. The negative control is cells transformed with the parental bait vector (pGBKT7) and prey vector (pACT2). (**B**) Schematic diagram displaying the identification strategy of IPMK-proximal/interacting proteins, which are biotinylated by APEX-tagged IPMK. (**C**) Bar graphs showing the relative abundance of biotinylated proteins related to the SWI/SNF complex and two negative controls (left). Target proteins were arranged according to their significance (left, significant; right, not significant). The volcano plot shows the relative abundance and significance (*P*-value) of biotinylated proteins related to the SWI/SNF complex and two negative controls (right). A dotted line within the volcano plot indicates p = 0.05. The relative abundance (abundance ratio) was derived by comparing the fold enrichment of target proteins in IPMK-APEX2-expressing vs. APEX2-expressing HEK293 cells. *P*-values were calculated using the Student's *t-test*, and error bars denote the standard error of the mean obtained from three biological replicates.

The online version of this article includes the following source data and figure supplement(s) for figure 1:

**Source data 1.** Full and unedited images corresponding to panel A.

**Source data 2.** Primary data for graphs in panel C.

**Figure supplement 1.** Various subunits of the SWI/SNF complex and histones are IPMK-proximal/interacting proteins.

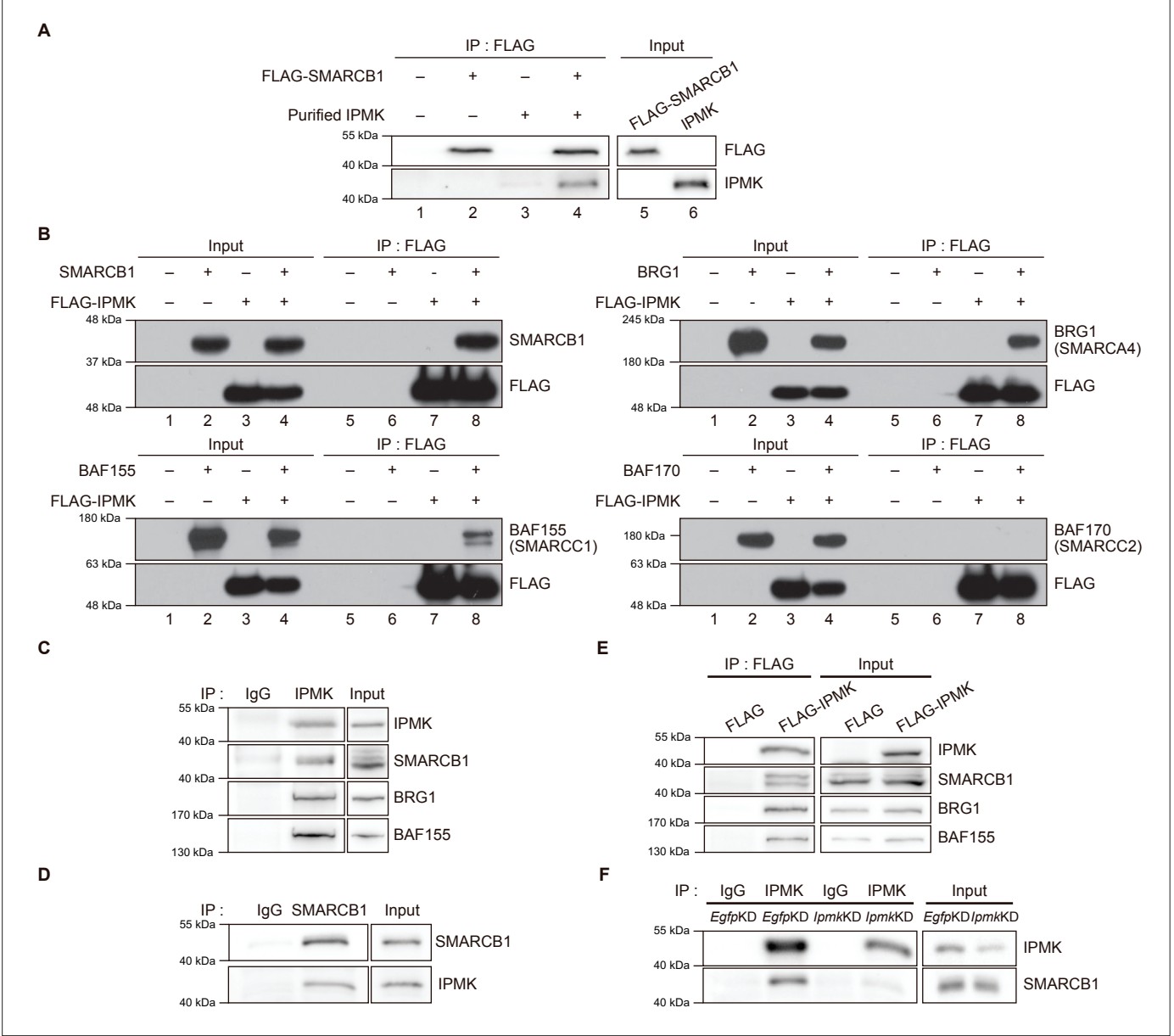

**Figure 2.** IPMK binds to SMARCB1 and other components of the SWI/SNF complex. (**A**) Purified IPMK and in vitro translated FLAG-SMARCB1 were co-incubated, immunoprecipitated with an anti-FLAG antibody, and then subjected to immunoblotting. (**B**) Sf9 insect cells were co-infected with baculoviruses expressing FLAG-IPMK and individual subunits of the SWI/SNF complex (SMARCB1, BRG1, BAF155, and BAF170), followed by FLAG M2 agarose immunoprecipitation and immunoblotting. (**C**) IPMK and IgG were immunoprecipitated from E14Tg2a cells and subjected to immunoblotting. (**D**) SMARCB1 and IgG were immunoprecipitated from E14Tg2a cells and subjected to immunoblotting. (**E**) E14Tg2a cells were transfected with FLAG-IPMK or FLAG (a control vector), followed by FLAG immunoprecipitation and immunoblotting. (**F**) E14Tg2a cells were transfected with siRNA against *Egfp* (*Egfp*KD) and *Ipmk* (*Ipmk*KD), immunoprecipitated with IPMK and IgG, and then subjected to immunoblotting.

The online version of this article includes the following source data and figure supplement(s) for figure 2:

**Source data 1.** Full and unedited blots corresponding to panel A.

**Source data 2.** Full and unedited blots corresponding to panel B.

**Source data 3.** Full and unedited blots corresponding to panel C.

**Source data 4.** Full and unedited blots corresponding to panel D.

**Source data 5.** Full and unedited blots corresponding to panel E.

**Source data 6.** Full and unedited blots corresponding to panel F.

**Figure supplement 1.** IPMK physically binds to SMARCB1 and other subunits of the SWI/SNF complex.

*Figure 2 continued on next page*

*Figure 2 continued*

**Figure supplement 1—source data 1.** Full and unedited blots corresponding to panel A.

**Figure supplement 1—source data 2.** Full and unedited blots corresponding to panel B.

**Figure supplement 1—source data 3.** Full and unedited blots corresponding to panel C.

**Figure supplement 1—source data 4.** Full and unedited blots corresponding to panel D.

**Figure supplement 1—source data 5.** Full and unedited blots corresponding to panel E.

**Figure supplement 1—source data 6.** Full and unedited blots corresponding to panel F.

**Figure supplement 1—source data 7.** Full and unedited blots corresponding to panel G.

**Figure supplement 1—source data 8.** Full and unedited blots and gels corresponding to panel H.

**Figure supplement 1—source data 9.** Full and unedited blots and gels corresponding to panel I.

vitro binding assays (*Figure 2A and B*), SMARCB1, BRG1, and BAF155 co-immunoprecipitated with endogenous IPMK (*Figure 2C*) and overexpressed FLAG-IPMK (*Figure 2E*) in mESC extracts. Intriguingly, we detected SMARCB1, BRG1, and BAF170 in the IPMK immunoprecipitates from MEF extracts (*Figure 2—figure supplement 1A, B*). To determine the specificity of the physical interaction between IPMK and SMARCB1, *Ipmk* and *Smarcb1* were knocked down (*Ipmk*KD and *Smarcb1*KD, respectively) in mESCs and MEFs. We confirmed the successful knockdown by quantifying the proteins (*Figure 2F*, *Figure 2—figure supplement 1D, E*). We observed that *Ipmk*KD did not affect the protein levels of the SWI/SNF complex subunits and *Smarcb1*KD did not affect the IPMK protein levels in mESCs or MEFs (*Figure 2—figure supplement 1D*). Importantly, a significant reduction in SMARCB1 signals was observed in the IPMK immunoprecipitate from *Ipmk*KD mESCs compared to control (*Egfp*KD) mESCs (*Figure 2F*). We also found that the SMARCB1 and BRG1 signals in IPMK immunoprecipitate from *Smarcb*KD MEFs were significantly decreased compared to signals in the control (*Egfp*KD) MEFs (*Figure 2—figure supplement 1E*). Furthermore, in SMARCB1 immunoprecipitate, we detected a significant reduction in IPMK signals in IPMK-null MEFs (*Figure 2—figure supplement 1C*). Taken together, these results indicate that IPMK directly associates with the core subunits of the SWI/SNF complex in vivo: IPMK-SMARCB1/BRG1/BAF155 in mESCs, IPMK-SMARCB1/BRG1/BAF170 in MEFs, and IPMK-SMARCB1 binding is specific and observed in both mESCs and MEFs.

Considering the above results, it is highly plausible that IPMK physically interacts with the SWI/SNF complex. To confirm this, we performed a co-immunoprecipitation assay by co-expressing IPMK and SMARCB1 in HEK293T cells. Consistent with the above results, we observed a physical interaction between IPMK and SMARCB1 (*Figure 2—figure supplement 1F*). Next, we performed GST pull-down assays by overexpressing GST-IPMK or GST alone (negative control) in HEK293T cells. Notably, the core subunits of the SWI/SNF (BAF)/PBAF complexes, namely SMARCB1, BAF155, BAF170, PBRM1, BAF250A, and BRM, were pulled down alongside GST-IPMK (*Figure 2—figure supplement 1G*), which was not seen in the negative control. Lastly, we purified endogenous SWI/SNF (BAF) complexes from cells expressing FLAG-DPF2 (*Figure 2—figure supplement 1H*). We co-incubated the purified SWI/SNF complexes with purified GST-IPMK or GST alone and performed GST pull-down assays. As expected, we detected the core subunits of the SWI/SNF complex (SMARCB1, BRG1, BAF155, and BAF170) in GST-IPMK pull-downs (*Figure 2—figure supplement 1I*). Collectively, our results strongly imply that IPMK physically interacts with the mammalian SWI/SNF complex by directly binding to SMARCB1, BRG1, and BAF155 (SMARCC1).

## Mapping the binding sites between IPMK and SMARCB1

Among the three IPMK-binding proteins (SMARCB1, BRG1, and BAF155), SMARCB1 exhibited the most robust interaction with IPMK (*Figures 1 and 2*). Therefore, we conducted yeast two-hybrid assays to identify the domains of SMARCB1 that are required for the interaction with IPMK. We constructed various prey vectors encoding different SMARCB1 domains for the two-hybrid analyses. Interestingly, the prey vectors expressing amino acids 99–245, 99–319, and full-length SMARCB1 resulted in positive signals in the two-hybrid system (*Figure 3—figure supplement 1A*), indicating that the Rpt1 and Rpt2 domains of SMARCB1 participate in the protein-protein interactions between SMARCB1 and IPMK.

To further dissect the reciprocal binding sites required for SMARCB1-IPMK binding, various SMARCB1 deletion constructs were designed and overexpressed in HEK293T cells. First, based on the domain map of SMARCB1 (*Figure 3A*, top), we deleted the C-terminus (*Figure 3A*, middle). We confirmed that the Rpt1 domain of SMARCB1 is essential for the IPMK interaction by immunoprecipitating the overexpressed deletion constructs (*Figure 3B*, compare lane 3 with lanes 2, 4, and 5). Next, we generated additional N-terminal deleted SMARCB1 constructs (*Figure 3A*, bottom). Consistent with the results of the yeast two-hybrid assays (*Figure 3—figure supplement 1A*), we observed that the constructs containing the Rpt1 or Rpt2 domains interact with IPMK (*Figure 3C*, lanes 2, 3, 4, and 5 show positive signals). By independently overexpressing each SMARCB1 domain, we found that both the Rpt1 and Rpt2 domains of SMARCB1 could bind to IPMK (*Figure 3D and E*, see lanes 3, 4, and 6). We also dissected Rpt1 and Rpt2 domains into β-sheets and α-helices based on their structures. In Rpt1, two β-sheets and two α-helices are required for IPMK binding (*Figure 3—figure supplement 1B*, compare lane 4 with lanes 2 and 3). In Rpt2, only two β-sheets and no α-helices were bound to IPMK (*Figure 3—figure supplement 1C*, compare lane 2 with lane 3). These observations both sheets and helices of Rpt1 and/or only sheets of Rpt2 are required for IPMK binding were supported by overexpressing combinations of Rpt1 and Rpt2 domains (*Figure 3—figure supplement 1D*, lanes 3 and 4 show positive signals). This was further confirmed by the SMARCB1 lacking Rpt1 and Rpt2 not being able to bind IPMK (*Figure 3—figure supplement 1E*, compare lane 2 with lane 3, and F). Thus, we concluded that the sheets/helices of Rpt1 and sheets of Rpt2 are the major IPMK-interacting sites in SMARCB1.

Reciprocally, to identify which IPMK domains are required for SMARCB1 binding, we designed several GST-tagged deletion constructs of IPMK (*Figure 3G*) and conducted immunoprecipitation experiments. IPMK-SMARCB1 binding was primarily mediated by three IPMK regions, including exon 3, exon 4, and exon 6 (*Figure 3F*, lanes 4, 5, 7, and 8 show positive signals, and *Figure 3G*), which comprise the inositol binding site and the kinase domain. Taken together, our results elucidated the specific reciprocal biding sites between IPMK and SMARCB1.

## Co-localization of IPMK and BRG1 on the genome

Our results demonstrated that IPMK binds directly to the core subunits of the mammalian SWI/SNF complex (SMARCB1, BRG1, and BAF155) and, thus, physically interacts with the SWI/SNF complex. Accordingly, we speculate that IPMK plays an important role in chromatin regulation. However, the cellular region where this IPMK-SWI/SNF interaction occurs in vivo and the detailed localization and role of IPMK in the chromatin remains elusive. Therefore, we conducted a chromatin fractionation assay using mESCs and MEFs. IPMK was evenly distributed in all the three subcellular fractions (cytoplasm, nucleoplasm, and chromatin), whereas SMARCB1 and BRG1 were primarily found in the chromatin fraction for both mESCs and MEFs (*Figure 4—figure supplement 1A-D*). To further investigate whether the expression of IPMK and SMARCB1 affects the distribution of the other, we conducted RNAi-mediated *Ipmk*KD and *Smarcb1*KD before the chromatin fractionation assay. We observed that the distribution of SMARCB1 was unaffected by *Ipmk*KD (*Figure 4—figure supplement 1B*), and the distribution of IPMK was unaffected by *Smarcb1*KD (*Figure 4—figure supplement 1D*). Taken together, these results indicate that IPMK, BRG1, and SMARCB1 reside together on the chromatin.

Next, we sought to determine where the IPMK, BRG1, and SMARCB1 localization takes place within the chromatin. To investigate the localization of BRG1 and IPMK within the chromatin, we performed cleavage under targets and release using nuclease (CUT&RUN) assays (*Skene and Henikoff, 2017*) in mESCs. In accordance with our results showing the physical association of IPMK and the SWI/SNF complex, we found that IPMK co-localizes with BRG1 on the genome (*Figure 4B*). Next, we performed peak annotation to analyze the genomic regions (e.g. promoters or intergenic regions) enriched with CUT&RUN peaks. Given that BRG1, a catalytic subunit of the SWI/SNF complex, is known to localize at the promoter-TSS (*de Dieuleveult et al., 2016*), we confirmed that BRG1 is significantly enriched at promoters in the mouse genome (*Figure 4E and F*). Notably, IPMK was also significantly enriched at promoters (*Figure 4E and F*). As the above peak annotation analysis did not specify the enhancers, which show a strong BRG1 enrichment (*de Dieuleveult et al., 2016*; *Laurette et al., 2015*; *Nakayama et al., 2017*), we employed the previously determined enhancer lists from mESCs (*Whyte et al., 2013*) and aligned them with BRG1 peaks. Consistent with previous findings, highly enriched BRG1 chromatin immunoprecipitation sequencing (ChIP-seq) peaks from the previous

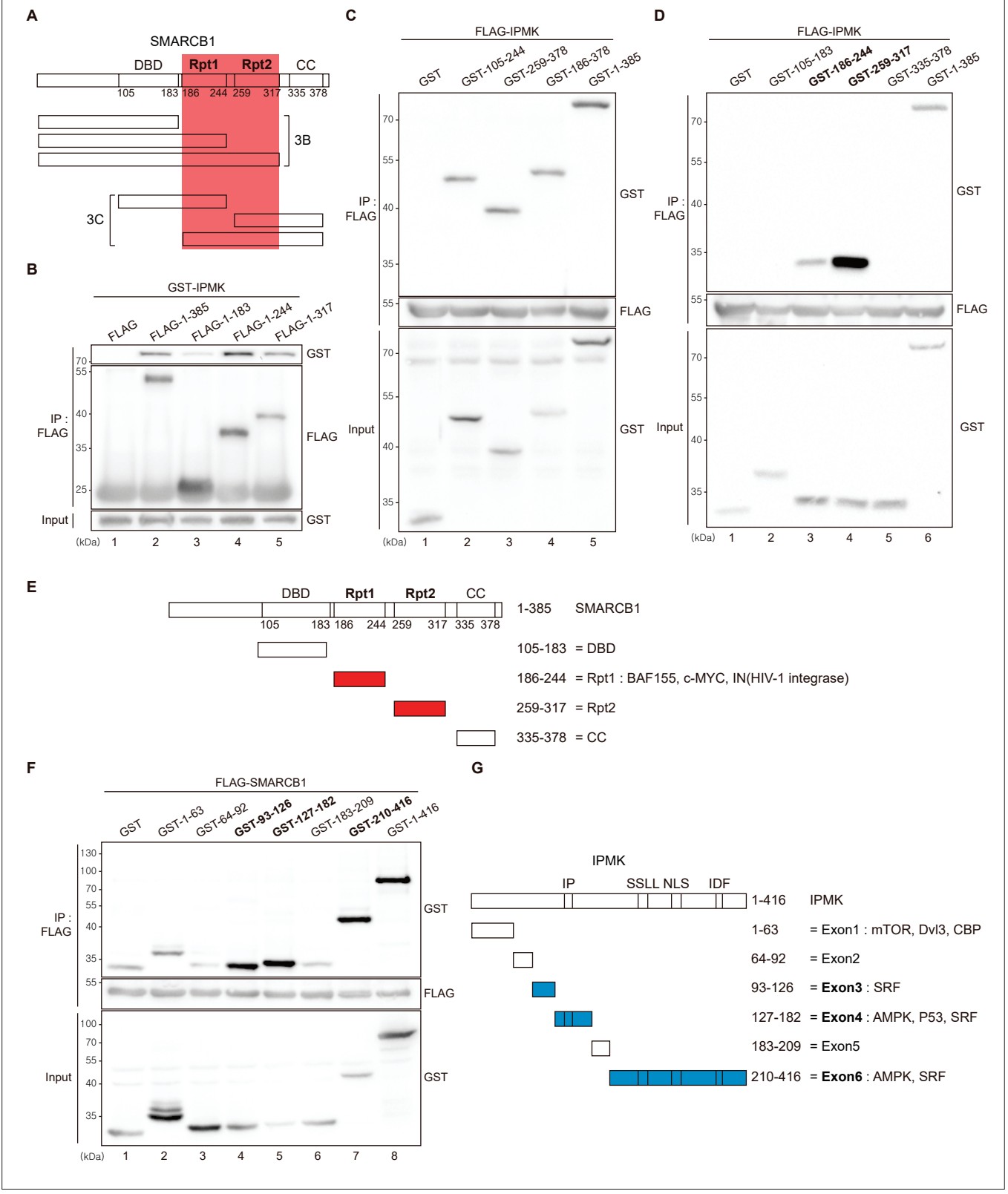

**Figure 3.** Identification of the distinct domains required for IPMK-SMARCB1 interactions. (**A**) Schematic diagram showing the human SMARCB1 fragments used for the binding studies (**B and C**). The IPMK-binding sites (Rpt1 and Rpt2) are highlighted in red. (**B**) HEK293T cells were co-transfected with GST-IPMK and FLAG (a control vector) or FLAG-SMARCB1 fragments, followed by immunoprecipitation with an anti-FLAG antibody, and then subjected to immunoblotting. (**C and D**) HEK293T cells were co-transfected with FLAG-IPMK and GST (a control vector) or GST-SMARCB1 fragments,

*Figure 3 continued on next page*

*Figure 3 continued*

followed by immunoprecipitation with an anti-FLAG antibody, and then subjected to immunoblotting. The specific IPMK-binding SMARCB1 fragments are in bold. (**E**) Schematic diagram showing the domains of human SMARCB1. SMARCB1 fragments used for the binding studies in (**D**) are indicated below with the numbers of amino acid sequences. The specific IPMK-binding SMARCB1 fragments (Rpt1 and Rpt2) are highlighted in red. (**F**) HEK293T cells were co-transfected with FLAG-SMARCB1 and GST (a control vector) or GST-IPMK fragments, followed by immunoprecipitation with an anti-FLAG antibody, and then subjected to immunoblotting. (**G**) Schematic diagram showing the domains of human IPMK. IPMK fragments used for the binding studies in (**F**) are indicated below with the numbers of amino acid sequences. Key domains for inositol binding (IP), kinase activity (SSLL and IDF), and the nuclear localization signal (NLS) are depicted. The specific SMARCB1-binding IPMK fragments (exons 3, 4, and 6) are highlighted in blue.

The online version of this article includes the following source data and figure supplement(s) for figure 3:

**Source data 1.** Full and unedited blots corresponding to panel B.

**Source data 2.** Full and unedited blots corresponding to panel C.

**Source data 3.** Full and unedited blots corresponding to panel D.

**Source data 4.** Full and unedited blots corresponding to panel F.

**Figure supplement 1.** Domain maps of the interaction between IPMK and SMARCB1.

**Figure supplement 1—source data 1.** Full and unedited images corresponding to panel A.

**Figure supplement 1—source data 2.** Full and unedited blots corresponding to panel B.

**Figure supplement 1—source data 3.** Full and unedited blots corresponding to panel C.

**Figure supplement 1—source data 4.** Full and unedited blots corresponding to panel D.

**Figure supplement 1—source data 5.** Full and unedited blots corresponding to panel E.

study (*Figure 4—figure supplement 1E*) and highly enriched BRG1 CUT&RUN peaks from our study (*Figure 4B and D*, and *Figure 4—figure supplement 1F*) mostly localized at enhancers (upper part of the heatmap). Furthermore, both BRG1 peaks (ChIP-seq peaks from the previous study and our CUT&RUN peaks) showed similar enrichment patterns in promoters and enhancers (*Figure 4—figure supplement 1G*), validating our CUT&RUN experiments. Notably, we found that highly enriched IPMK also localized at enhancers (*Figure 4B*).

Collectively, these results strongly indicate that IPMK and BRG1 co-localize at the chromatin, particularly in the promoter and enhancer regions, which further supports our previous results of a physical association between IPMK and the SWI/SNF complex.

## IPMK regulates the genomic occupancy of BRG1 and BRG1-mediated chromatin accessibility

To further elucidate the role of IPMK in BRG1 localization, we performed BRG1 CUT&RUN assays upon *Ipmk*KD (*Figure 4A*) and compared them to *Egfp*KD (control) in mESCs. Interestingly, we detected a significant reduction in genome-wide BRG1 intensity upon *Ipmk*KD at BRG1 CUT&RUN peaks, with low BRG1 enrichment in *Egfp*KD mESCs (bottom half of the heatmaps, termed as Low; *Figure 4B* and *Figure 4—figure supplement 2A*). To confirm this reduction in BRG1, we established constitutive IPMK-depleted mESCs by shRNA-mediated *Ipmk* silencing (*Figure 4C* and *Figure 4—figure supplement 2C*) and performed BRG1 CUT&RUN assays upon IPMK depletion (sh*Ipmk1* and sh*Ipmk2*), comparing them to shNT (a non-target control). Consistent with the decreased BRG1 intensity shown in siRNA-mediated *Ipmk* silencing (*Figure 4B*), we detected a significant reduction in genome-wide BRG1 intensity upon shRNA-mediated *Ipmk* silencing at BRG1 CUT&RUN peaks with low BRG1 enrichment in shNT mESCs (bottom half of the heatmaps, termed as Low; *Figure 4D*). Although some differences exist (number of BRG1 peaks and their positions) between BRG1 CUT&RUN of siRNA- and shRNA-mediated *Ipmk* silencing, we confirmed the reduced BRG1 intensity upon IPMK depletion at the Low BRG1 CUT&RUN merged peaks (merged peaks derived by merging BRG1 CUT&RUN peaks of *Egfp*KD and shNT mESCs) in both siRNA and shRNA experiments (*Figure 4—figure supplement 2D and E*). The observed reduction in BRG1 intensity at Low BRG1 peaks was not derived from any changes in the expression of BRG1 or other subunits of the SWI/SNF complex (*Figure 2—figure supplement 1D Figure 4—figure supplement 1B*, *Figure 4—figure supplement 2B and C*), excluding the possibility that IPMK depletion results in reduced expression of BRG1 or another SWI/SNF complex subunit that leads to reduced BRG1 occupancy. In addition, the genomic distribution of BRG1 CUT&RUN peaks was unaffected by *Ipmk*KD (*Figure 4E and F*), suggesting that IPMK depletion

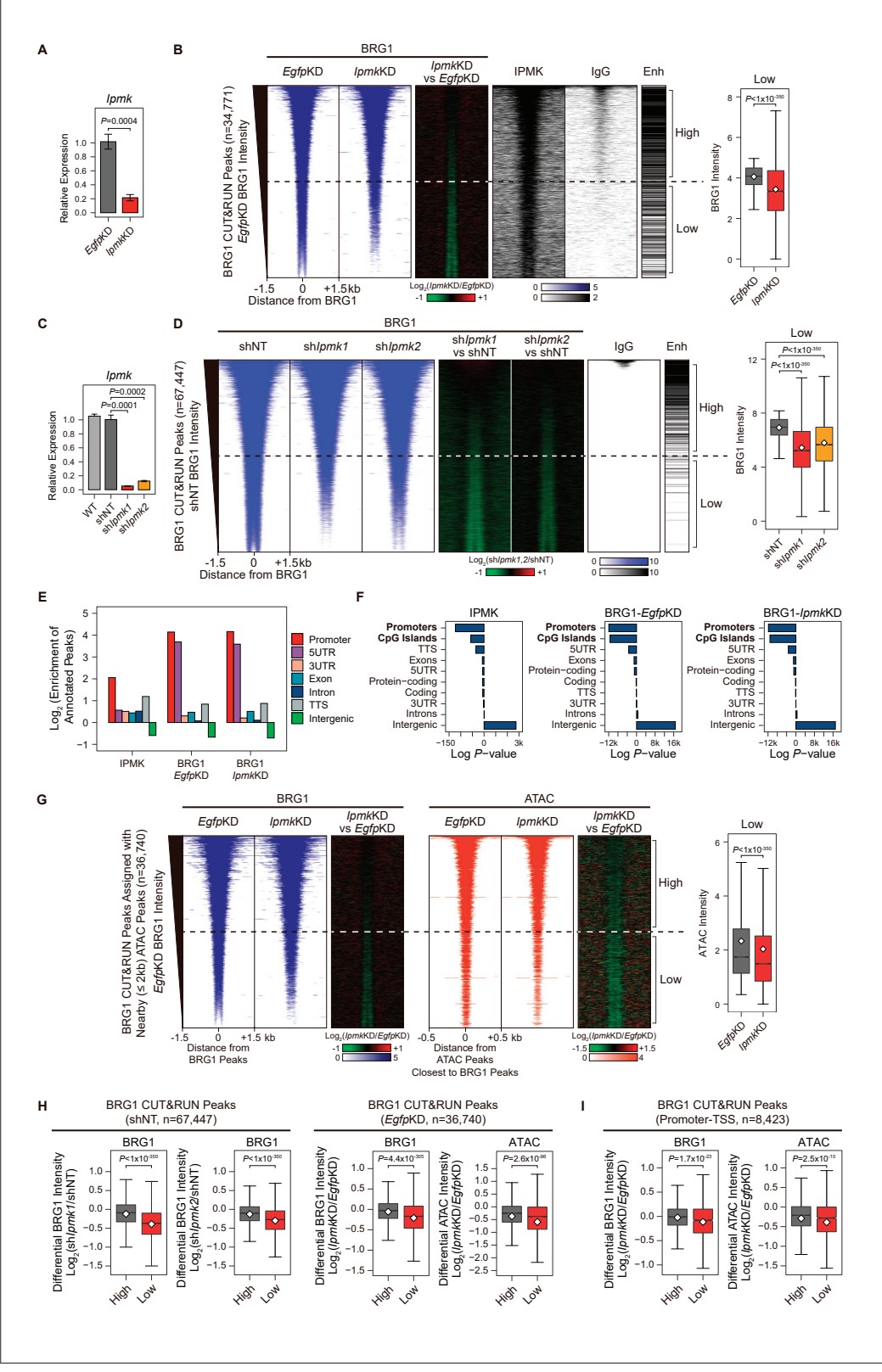

**Figure 4.** IPMK-BRG1 co-localize at promoters-TSSs/enhancers, and IPMK regulates the genomic localization of BRG1. (**A**) RT-qPCR analysis of *Ipmk* expression after siRNA treatment. Error bars denote the standard error of the mean obtained from four biological replicates. The expression levels were normalized to the expression of β-actin. *P*-values were calculated using the Student's *t-test*. (**B**) Heatmaps representing CUT&RUN results for BRG1

*Figure 4 continued on next page*

*Figure 4 continued*

(*Egfp*KD, *Ipmk*KD, and their comparison), IPMK, and IgG, and enhancer (Enh) distribution at BRG1 CUT&RUN peaks (*Egfp*KD cells) as indicated at the top (left). All heatmaps were aligned at 34,771 BRG1 CUT&RUN peaks (rows) and sorted in descending order by the BRG1 intensity of *Egfp*KD cells. High and Low groups were divided equally (n = 17,385 and 17,386, respectively) according to the BRG1 intensity of *Egfp*KD cells. Box plots show the BRG1 intensity at Low BRG1 CUT&RUN peaks (right). (**C**) RT-qPCR analysis of *Ipmk* expression in wild-type mESCs (WT) and after shRNA-mediated *Ipmk* silencing. Error bars denote the standard error of the mean obtained from four biological replicates. The expression levels were normalized to the expression of *β-actin*. *P*-values were calculated using the Student's *t-test*. (**D**) Heatmaps representing CUT&RUN results for BRG1 (shNT, sh*Ipmk1*, sh*Ipmk2*, and their comparison) and IgG, and enhancer (Enh) distribution at BRG1 CUT&RUN peaks (shNT cells) as indicated at the top (left). All heatmaps were aligned at 67,447 BRG1 CUT&RUN peaks (rows) and sorted in descending order by the BRG1 intensity of shNT cells. High and Low groups were divided equally (n = 33,723 and 33,724, respectively) according to the BRG1 intensity of shNT cells. Box plots show the BRG1 intensity at the Low BRG1 CUT&RUN peaks (right). (**E**) Bar graphs showing the Log$_2$ enrichment of CUT&RUN peaks (IPMK and BRG1-*Egfp*KD, -*Ipmk*KD) annotated with various regions of the mouse genome. (**F**) Bar graphs showing the significance (Log *P*-value) of CUT&RUN peaks (IPMK and BRG1-*Egfp*KD, -*Ipmk*KD) annotated with various regions of the mouse genome. For each CUT&RUN peak, genome annotations (e.g., promoters or CpG islands) are sorted in descending order according to their significance (*P*-values, decreasing significance from the top toward the bottom). (**G**) Heatmaps representing BRG1 CUT&RUN (*Egfp*KD, *Ipmk*KD, and their comparison) at BRG1 CUT&RUN peaks (*Egfp*KD cells) assigned with nearby (within 2 kb) ATAC-seq peaks (left). BRG1 peaks without nearby ATAC-seq peaks were excluded. To match the arrangement with ATAC-seq peaks (right), a BRG1 peak containing multiple ATAC-seq peaks was included without deduplication. Heatmaps representing ATAC-seq signals (*Egfp*KD, *Ipmk*KD, and their comparison) at ATAC-seq peaks assigned with the closest BRG1 CUT&RUN peaks that were used for heatmaps on the left (right). All heatmaps were aligned at 36,740 BRG1 CUT&RUN peaks (left) or 36,740 ATAC-seq peaks (right) and sorted in descending order by the BRG1 intensity of *Egfp*KD cells. High and Low groups were divided equally (n = 18,370 and 18,370, respectively) according to the BRG1 intensity of *Egfp*KD cells. Box plots show the BRG1 intensity at the Low BRG1 CUT&RUN peaks (right). (**H**) Box plots showing the differential BRG1 and ATAC intensity upon IPMK depletion at the High (grey) and Low (red) BRG1 CUT&RUN peaks and corresponding (closest) ATAC-seq peaks. High and Low groups were divided according to the BRG1 intensity of shNT (left) or *Egfp*KD (right) cells. (**I**) Box plots showing the differential BRG1 (left) and ATAC (right) intensity upon *Ipmk*KD at the High (grey) and Low (red) BRG1 CUT&RUN peaks localized at promoters-TSSs (left) and corresponding (closest) ATAC-seq peaks (right). High and Low groups (n = 5640 and 2783, respectively) were derived from (**G**). (**B, D, G, H and I**) *P*-values were calculated using the Wilcoxon rank sum test.

The online version of this article includes the following source data and figure supplement(s) for figure 4:

**Source data 1.** Primary data for graph in panel A.

**Source data 2.** Primary data for graph in panel C.

**Figure supplement 1.** Chromatin fraction assay and BRG1 at enhancers.

**Figure supplement 1—source data 1.** Full and unedited blots corresponding to panel A.

**Figure supplement 1—source data 2.** Full and unedited blots corresponding to panel B.

**Figure supplement 1—source data 3.** Full and unedited blots corresponding to panel C.

**Figure supplement 1—source data 4.** Full and unedited blots corresponding to panel D.

**Figure supplement 2.** mRNA expression of SWI/SNF complex subunits, replicates of BRG1 CUT&RUN and ATAC-seq, and IPMK rescue experiments.

**Figure supplement 2—source data 1.** Full and unedited blots corresponding to panel C.

**Figure supplement 2—source data 2.** Full and unedited blots corresponding to panel F.

does not affect the global distribution (changes in peak positions) of BRG1, but impacts the global occupancy of BRG1, especially at the low-enriched BRG1 peaks (Low), which do not correlate with enhancers (*Figure 4B and D*).

To address whether the enzymatic activity of IPMK is required for regulation of BRG1 occupancy, we performed rescue experiments by transfecting DNA constructs of wild-type (WT) IPMK (+WT) or catalytically dead IPMK (+SA) in *Ipmk*KD mESCs. We confirmed the stable expression of WT and catalytically dead IPMK (*Figure 4—figure supplement 2F*) and then performed BRG1 CUT&RUN assays. Consistent with our previous results (*Figure 4B and D*), we observed significantly reduced BRG1 intensity at Low BRG1 peaks upon *Ipmk*KD (*Figure 4—figure supplement 2G and H*). Although +WT transfection did not fully restore the BRG1 intensity, we observed a significantly increased BRG1

intensity in *Ipmk*KD +WT compared to *Ipmk*KD mESCs (*Figure 4—figure supplement 2G and H*). Interestingly, +SA transfection also exhibited significantly increased BRG1 intensity compared to *Ipmk*KD mESCs and did not show a large difference in BRG1 intensity compared to *Ipmk*KD +WT mESCs (*Figure 4—figure supplement 2G and H*). These results indicate that the enzymatic activity of IPMK is not required for the regulation of BRG1 occupancy in mESCs.

BRG1 was previously shown to regulate chromatin accessibility at nucleosome free regions (NFRs) of TSS in mESCs (*de Dieuleveult et al., 2016*). To investigate the effect of the *Ipmk*KD-induced decrease in BRG1 occupancy on chromatin accessibility, we performed an assay for transposase-accessible chromatin using sequencing (ATAC-seq) upon *Ipmk*KD and compared the results to *Egfp*KD (control) in mESCs. To precisely assess the effect of an *Ipmk*KD-induced decrease in BRG1 occupancy on chromatin accessibility, we assigned ATAC peaks to the nearby (within 2 kb) BRG1 peaks and selected these BRG1 peaks for further analysis; we excluded BRG1 peaks without nearby ATAC peaks (*Figure 4G*). In addition, BRG1 peaks containing or assigned multiple ATAC peaks were included without de-duplication to match the same ordering as ATAC peaks (the same alignment of heatmap rows was applied for BRG1 and ATAC peaks in *Figure 4G*). As expected, we observed that the global BRG1 occupancy was reduced upon *Ipmk*KD at BRG1 peaks with low BRG1 intensity (Low; *Figure 4G*), consistent with our previous observations (*Figure 4B*). Importantly, at Low BRG1 peaks, both BRG1 occupancy and BRG1-mediated chromatin accessibility (ATAC-seq signals closest to the BRG1 CUT&RUN peaks) were significantly reduced upon *Ipmk*KD in a genome-wide manner (*Figure 4G* and *Figure 4—figure supplement 2I*). Notably, when comparing the highly enriched (High) and lowly enriched (Low) BRG1 peaks, IPMK depletion greatly reduced the BRG1 and ATAC-seq intensity, especially at Low BRG1 peaks (*Figure 4H*). Reduced BRG1 and BRG1-associated chromatin accessibility (ATAC-seq) upon *Ipmk*KD were also detected Low BRG1 peaks residing at promoter-TSS regions (*Figure 4I*), consistent with the previous study in which BRG1 primarily maintained chromatin accessibility at promoter-TSS regions (*de Dieuleveult et al., 2016*). Taken together, these results indicate that IPMK regulates the global BRG1 occupancy and corresponding BRG1-mediated chromatin accessibility in mESCs.

## IPMK affects BRG1 localization and chromatin accessibility at promoter-TSSs

As IPMK depletion greatly affects the BRG1 occupancy and ATAC-seq intensity at Low BRG1 peaks (*Figure 4G–I*), where enhancers are sparse (*Figure 4B and D*), we focused on the promoter-TSS regions, where BRG1 and IPMK were significantly enriched (*Figure 4E and F*). To examine the promoter-TSSs that are affected by the *Ipmk*KD-driven global loss of BRG1 occupancy, we initially plotted BRG1, nucleosome, and ATAC-seq signals at the TSS by analyzing BRG1 CUT&RUN, MNase-seq (chromatin digestion with micrococcal nuclease combined with sequencing), and ATAC-seq in *Egfp*KD mESCs. We confirmed that BRG1 is abundant near TSSs and observed two major BRG1 peaks, which were divided by the TSS and coincided with the −1 and +1 nucleosomes (*Figure 5A*, left). Furthermore, we observed the nucleosome-depleted/chromatin accessible regions, known as NFRs, which is a hallmark of the TSS (*Figure 5A*, left). To quantitatively assess the BRG1 and ATAC-seq signals, we defined three genomic regions and used them for subsequent analysis: Upstream and Downstream regions that coincided with the two major BRG1 peaks and Center regions that coincided with the ATAC-seq peaks (which also coincided with NFRs; *Figure 5A*). Next, using the signal intensity of ChIP-seq against histone H3K4me3 and H3K27me3, we classified promoters into three types: H3K4me3-Low, H3K4me3-Only (high H3K4me3, low H3K27me3), and bivalent (high H3K4me3, high H3K27me3). Consistent with the previous study (*de Dieuleveult et al., 2016*), both BRG1 intensity and chromatin accessibility were at high levels in H3K4me3-Only, moderate levels in bivalent, and low levels in H3K4me3-Low promoters (*Figure 5A*, right), confirming our promoter classification method. To dissect 9,042 TSSs with decreased BRG1 occupancy upon *Ipmk*KD (1.5-fold changes in BRG1 occupancy compared to *Egfp*KD), we categorized them into the three promoter types (*Figure 5B*) and five clusters based on the combinatorial changes in BRG1 levels upon *Ipmk*KD at the previously defined Up and Downstream regions (*Figure 5C*). We excluded the H3K4me3-Low promoters, which exhibited extremely low BRG1 signals (*Figure 5A*, top right). H3K4me3-Only promoters (*Figure 5B and C*) and Cluster2/3 (*Figure 5C*) occupied a large proportion of promoter-TSSs with decreased BRG1 levels upon *Ipmk*KD. Notably, BRG1 occupancy was reduced upon *Ipmk*KD at both promoters and five distinctive clusters in accordance with our predefined classification (*Figure 5D and E*).

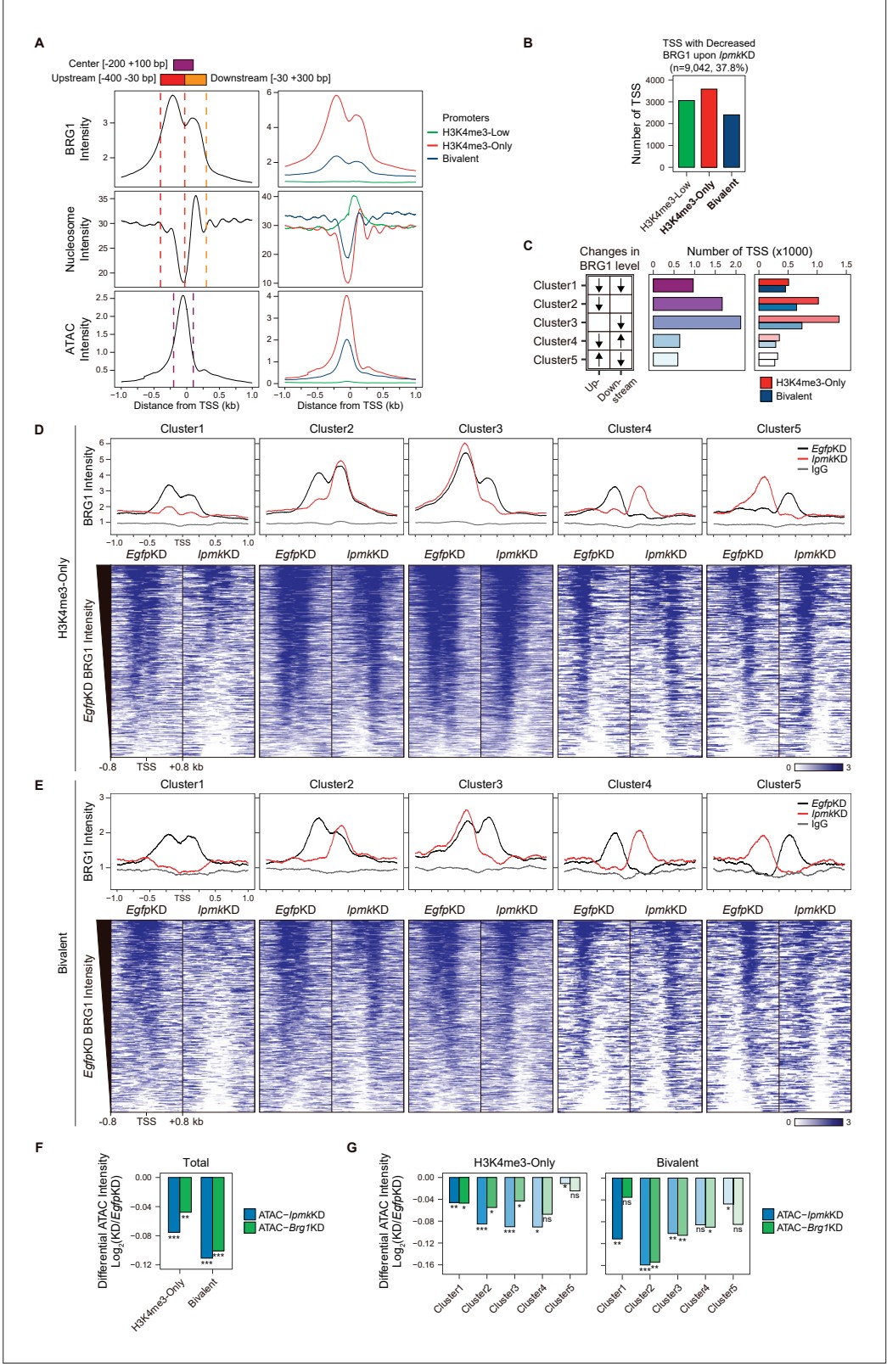

**Figure 5.** IPMK affects chromatin accessibility at promoter-TSS by regulating BRG1 localization. (**A**) Line plots showing the average enrichment of BRG1, nucleosome (MNase-seq, GSM5253962, and GSM5253963), and ATAC-seq signals (ATAC) at the TSSs of total genes (left) and TSSs of three promoter types (right). Three genomic regions, indicated at the top (see also dotted lines on the left), were defined according to the relative position of

*Figure 5 continued on next page*

*Figure 5 continued*

enriched ATAC-seq intensity (Center, purple) and enriched BRG1 intensity (Upstream and Downstream, red and orange, respectively). Green, red, and blue lines on the right indicate H3K4me3-Low, H3K4me3-Only, and bivalent promoters, respectively. (**B**) Bar graphs showing the number of TSSs exhibiting decreased BRG1 intensity upon *Ipmk*KD with different promoter types. (**C**) A diagram displaying five clusters of TSSs classified by changes in the BRG1 level at Up/Downstream regions defined in (**A**) upon *Ipmk*KD (left). Downward and upward arrows indicate decreased and increased BRG1 levels upon *Ipmk*KD, respectively. Bar graphs showing the number of five TSS clusters (middle) with different promoter types (right). (**D**) Line plots showing the average enrichment of BRG1 (*Egfp*KD and *Ipmk*KD) and IgG at five TSS clusters with H3K4me3-Only promoters (top). Heatmaps representing BRG1 intensity (*Egfp*KD and *Ipmk*KD) at five TSS clusters as indicated at the top (bottom). (**E**) Line plots showing the average enrichment of BRG1 (*Egfp*KD and *Ipmk*KD) and IgG at five TSS clusters with bivalent promoters (top). Heatmaps representing BRG1 intensity (*Egfp*KD and *Ipmk*KD) at five TSS clusters as indicated at the top (bottom). (**D and E**) Black, red, and gray lines indicate BRG1 intensity with *Egfp*KD, *Ipmk*KD, and IgG intensity, respectively. All heatmaps are shown in descending order by the BRG1 intensity of *Egfp*KD cells. (**F**) Bar graphs showing the average differential ATAC-seq intensity ($Log_2$ KD/*Egfp*KD) upon *Ipmk*KD (blue) and *Brg1*KD (green) at the TSSs of two promoter types. (**G**) Bar graphs showing the average differential ATAC-seq intensity ($Log_2$ KD/*Egfp*KD) upon *Ipmk*KD (blue) and *Brg1*KD (green) at five TSS clusters with H3K4me3-Only (left) and bivalent promoters (right). (**F and G**) *p < 0.01, **p < 1 × $10^{-4}$, ***p < 1 × $10^{-10}$; ns, not significant; Wilcoxon signed rank test.

The online version of this article includes the following figure supplement(s) for figure 5:

**Figure supplement 1.** IPMK plays an important role in the maintenance of chromatin accessibility at promoter-TSS by regulating BRG1 localization.

Previously, BRG1 was reported to differentially regulate chromatin accessibility depending on the promoter types and where it localizes near TSSs in mESCs; BRG1 localized at the –1 nucleosome in a wide NFR (median length 808 bp) of H3K4me3-Only/bivalent promoters positively regulates the chromatin accessibility at the NFR, whereas BRG1 localized at the +1 nucleosome in a narrow NFR (median length 28 bp) of H3K4me3-Only promoters tends to inhibit chromatin accessibility at the NFR (**de Dieuleveult et al., 2016**). To elucidate the effect of an *Ipmk*KD-induced decrease in BRG1 occupancy on chromatin accessibility at two promoter types and five clusters, we analyzed two ATAC-seq data sets: ours (*Ipmk*KD) and one that was publicly released (*Brg1*KD, GSE64825). We calculated the differential ATAC-seq signals (KD vs. controls) at predefined Center regions. Consistent with our previous genome-wide results (**Figure 4G–I**), ATAC-seq signals were significantly reduced upon *Ipmk*KD at both promoter types (**Figure 5F**, Total of all clusters) and most clusters (**Figure 5G**). Notably, the ATAC-seq signals decreased similarly upon *Ipmk*KD and *Brg1*KD at two promoter types (**Figure 5F**) and five clusters (**Figure 5G**). As the reduced BRG1 occupancy upon *Ipmk*KD partially mimics *Brg1*KD, the similar result upon *Ipmk*KD and *Brg1*KD further supports IPMK playing a vital role in chromatin accessibility at the promoter-TSS by regulating the BRG1 occupancy. Intriguingly, we found that the ATAC-seq signals at bivalent promoters were reduced more upon *Ipmk*KD than those at H3K4me3-Only promoters (**Figure 5F and G**). Supporting this, we found that shRNA-mediated *Ipmk* silencing significantly reduced the BRG1 intensity only at bivalent promoters (**Figure 5—figure supplement 1A**) and observed a close association between a reduced BRG1 level upon IPMK depletion and decreased ATAC-seq signals upon *Ipmk*KD and *Brg1*KD at specific loci of bivalent promoters (**Figure 5—figure supplement 1B**). However, why *Ipmk*KD shows differential effects on chromatin accessibility at two promoter types is unclear; for example, although IPMK depletion reduced BRG1 occupancy at both promoters in Cluster1, why does it impact the chromatin accessibility of the bivalent promoters more than H3K4me3-Only promoters? To examine this, we compared the BRG1 occupancy in *Ipmk*KD cells (**Figure 5D and E**, red lines) at Cluster1 in two promoter types. H3K4me3-Only promoters contained more BRG1 occupancy than bivalent promoters (**Figure 5D and E**). In other words, although Cluster1 exhibited total BRG1 loss in *Ipmk*KD cells, some degree of BRG1 occupancy at H3K4me3-Only promoters remained, with higher levels than bivalent promoters (**Figure 5D and E**). For direct comparisons, we plotted BRG1 in *Ipmk*KD cells at two promoter types. H3K4me3-Only promoters had higher BRG1 occupancy compared to bivalent promoters at all clusters (**Figure 5—figure supplement 1C**). Thus, the remaining BRG1 at H3K4me3-Only promoters may maintain the chromatin accessibility, causing the differential effect between two promoter types. (**de Dieuleveult et al., 2016**) previously reported that the majority of chromatin remodelers (BRG1, CHD1, CHD2, CHD4, CHD6, CHD8, CHD9, and EP400) exhibit higher enrichment at H3K4me3-Only promoters

compared to bivalent promoters. To test this in our study, we plotted various chromatin remodelers at two promoter types in five clusters using the publicly released data sets (GSE64825). Consistently, we detected higher occupancy at H3K4me3-Only promoters compared to bivalent promoters for all of the chromatin remodelers (CHD1, CHD2, CHD4, CHD6, CHD8, CHD9, and EP400; *Figure 5—figure supplement 1D*). Thus, these highly enriched chromatin remodelers at H3K4me3-Only promoters may compensate for the loss of BRG1 upon IPMK depletion, thereby maintaining the chromatin accessibility. BRG1 depletion has also been reported to reduce the ATAC-seq (chromatin accessibility) signals at bivalent promoters in mESCs (*de Dieuleveult et al., 2016*); Brg1 depletion, which would cause total BRG1 loss at H3K4me3-Only and bivalent promoters similar to Cluster1 in our study, presented a greater reduction in ATAC-seq signals at bivalent promoters than H3K4me3-Only promoters, consistent with our results. Taken together, these results support differential chromatin accessibility at two promoter types in terms of an *Ipmk*KD-driven reduction in BRG1 occupancy.

The ATAC-seq signals at Cluster2 of bivalent promoters were reduced more than other clusters upon *Ipmk*KD (*Figure 5G*), which is consistent with the fact that, in bivalent promoters, BRG1 is localized at the –1 nucleosome and maintains chromatin accessibility (*de Dieuleveult et al., 2016*). Although H3K4me3-Only promoters also contain Cluster2, we did not detect a robust decrease in ATAC-seq signals at Cluster2 of H3K4me3-Only promoters (*Figure 5G*). This discrepancy between bivalent and H3K4me3-Only promoters may be due to the remaining BRG1 at H3K4me3-Only promoters (*Figure 5—figure supplement 1C*) or high enrichment of various chromatin remodelers at H3K4me3-Only promoters (*Figure 5—figure supplement 1D*). In addition, the different BRG1 occupancy in Upstream (–1 nucleosome) and Downstream ( + 1 nucleosome) regions of Cluster2 in *Egfp*KD mESCs may cause this discrepancy; BRG1 is highly enriched at Upstream regions compared to Downstream regions in bivalent promoters (*Figure 5E*, Cluster2, black line), whereas BRG1 levels are relatively similar at both Upstream and Downstream regions in H3K4me3-Only promoters (*Figure 5D*, Cluster2, black line). Interestingly, although we applied the same criteria when categorizing the five clusters, H3K4me3-Only and bivalent promoters exhibited different BRG1 localizations in *Egfp*KD mESCs at Cluster2 and Cluster3 (*Figure 5D and E*), indicating that BRG1 has distinct localization in these two promoter types in mESCs. Taken together, these results suggest that IPMK plays a pivotal role in maintaining the chromatin accessibility of bivalent promoters, particularly by safeguarding BRG1 occupancy at the –1 nucleosome.

Collectively, these findings indicate that IPMK regulates BRG1 occupancy and BRG1-mediated chromatin accessibility at promoter-TSS regions and suggest that IPMK depletion causes the most severe impacts on chromatin accessibility at bivalent promoters, which are strongly associated with the *Ipmk*KD-induced decrease in BRG1 occupancy at –1 nucleosomes.

## Loss of IPMK partially affects transcription by disrupting BRG1 localization and chromatin accessibility at promoter-TSSs

To investigate the effect of *Ipmk*KD-driven disruption of BRG1 occupancy and chromatin accessibility on transcription, we performed high-throughput mRNA sequencing (mRNA-seq) using mESCs with *Ipmk*KD. We calculated the differential mRNA expression (*Ipmk*KD vs. *Egfp*KD) of genes having promoters with decreased BRG1/ATAC-seq intensity upon *Ipmk*KD (*Figure 5D–F*, Total). The mRNA levels of genes with either promoter type were down-regulated upon *Ipmk*KD (*Figure 5—figure supplement 1E*). Intriguingly, we detected significant down-regulation of the mRNA levels of genes with bivalent promoters upon *Ipmk*KD compared to H3K4me3-Only promoters (*Figure 5—figure supplement 1E*). This discrepancy may arise from the greater impact of *Ipmk*KD on bivalent promoter-TSS chromatin accessibility than that of H3K4me3-Only promoters (*Figure 5F and G*, and *Figure 5—figure supplement 1A and B*). To check the decreased BRG1/ATAC-seq-associated down-regulation of gene expression upon *Ipmk*KD, we performed reverse transcription–quantitative polymerase chain reaction (RT-qPCR), a conventional method for checking the gene expression. We confirmed that mRNA levels of genes exhibiting decreased BRG1 occupancy upon *Ipmk*KD and reduced ATAC-seq signals upon *Ipmk*KD/*Brg1*KD at bivalent (*Figure 6A and B*) and H3K4me3-Only promoters (*Figure 6—figure supplement 1A*) were significantly down-regulated upon *Ipmk*KD (*Figure 6A and B*, and *Figure 6—figure supplement 1A*). To examine this in a genome-wide manner, we identified differentially expressed genes (DEGs) by comparing the gene expression in *Ipmk*KD cells to the expression in control (*Egfp*KD) cells and identified 300 DEGs that were down-regulated

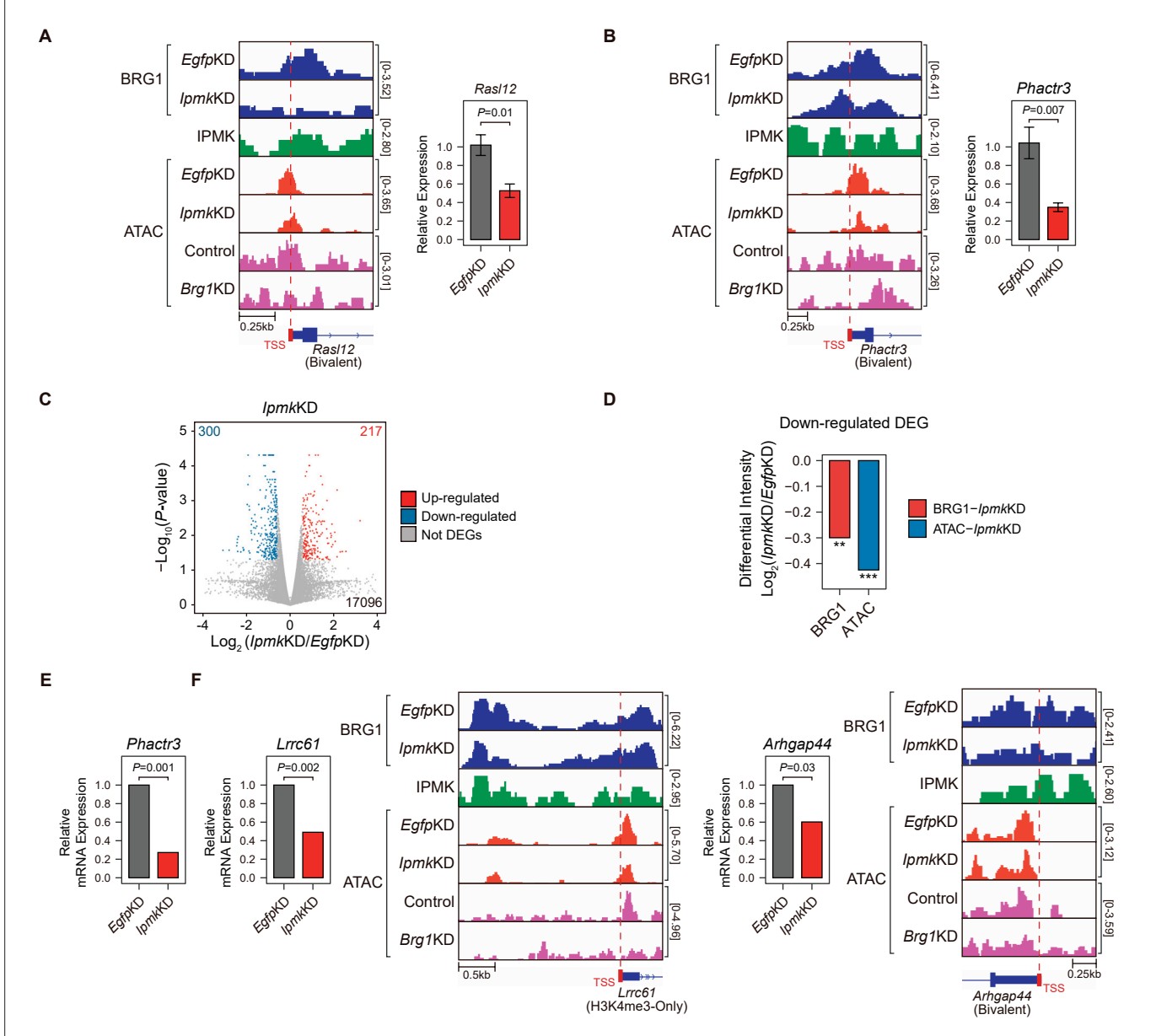

**Figure 6.** Alterations in BRG1/ATAC upon *Ipmk*KD affect gene expression. (**A and B**) Examples of BRG1 (*Egfp*KD and *Ipmk*KD), IPMK CUT&RUN, and ATAC-seq (*Egfp*KD, *Ipmk*KD, Control, and *Brg1*KD) assays at the TSSs of *Rasl12* (**A**) and *Phactr3* (**B**). The TSSs are marked with red boxes (bottom) and dotted lines. The graphs indicate RT-qPCR analysis of *Rasl12* (**A**) and *Phactr3* (**B**) expression after siRNA treatment. Error bars denote the standard error of the mean obtained from four biological replicates. The expression levels were normalized to the expression of *β-actin*. *P*-values were calculated using the Student's *t-test*. (**C**) Volcano plots showing the differentially expressed genes (DEGs) upon *Ipmk*KD based on the mRNA-seq data. Red and blue dots indicate significant up- and down-regulation, respectively (p ≤ 0.05 and fold change ≥1.5). (**D**) Bar graphs showing the average differential BRG1 (red) and ATAC-seq (blue) intensity upon *Ipmk*KD at BRG1 CUT&RUN peaks (for BRG1 intensity) and ATAC-seq peaks (for ATAC intensity) that are closest (within 2 kb for BRG1 peaks and within 500 bp for ATAC-seq peaks) to the TSSs of the down-regulated DEGs. **p < 1 × 10⁻⁵, ***p < 1 × 10⁻¹⁰, Wilcoxon signed rank test. (**E**) mRNA-seq analysis of *Phactr3* expression after siRNA treatment. *P*-values were calculated using the Student's *t-test*. (**F**) Examples of BRG1 (*Egfp*KD and *Ipmk*KD), IPMK CUT&RUN, and ATAC-seq (*Egfp*KD, *Ipmk*KD, Control, and *Brg1*KD) assays at the TSSs of *Lrrc61* (left) and *Arhgap44* (right). The BRG1 CUT&RUN peaks (*Egfp*KD cells) and ATAC-seq peaks are marked at the top with blue and green boxes, respectively. TSSs are marked with red boxes (bottom) and dotted lines. mRNA-seq analysis of *Lrrc61* (left) and *Arhgap44* (right) expression after siRNA treatment. *P*-values were calculated using the Student's *t-test*.

The online version of this article includes the following source data and figure supplement(s) for figure 6:

**Source data 1.** Primary data for graph in panel A.

*Figure 6 continued on next page*

*Figure 6 continued*

**Source data 2.** Primary data for graph in panel B.

**Source data 3.** Primary data for graph in panel E.

**Source data 4.** Primary data for graphs in panel F.

**Figure supplement 1.** Alterations in BRG1/ATAC upon *Ipmk*KD affect gene expression.

**Figure supplement 1—source data 1.** Primary data for graphs in panel A.

---

(*Figure 6C*). Notably, we observed that BRG1 occupancy and BRG1-mediated chromatin accessibility (ATAC-seq signals) were both significantly reduced near the TSSs of these down-regulated genes (*Figure 6D*). We confirmed this finding by monitoring the specific gene loci, including *Phactr3*, *Lrrc61*, and *Arhgap44* (*Figure 6B, E and F*). Thus, in mESCs, IPMK maintains the expression of a subset of genes by safeguarding the appropriate BRG1 occupancy and BRG1-mediated chromatin accessibility at the corresponding promoter-TSSs.

## IPMK regulates the expression of endodermal marker genes

To investigate the significance of IPMK function in mESC biology, we employed stable shRNA-mediated IPMK depletion and measured the differentiation kinetics by aggregating mESCs into embryoid bodies (EBs), which are in vitro cell clumps that recapitulate the early events of embryogenesis. We harvested the mESCs at various time points during EB formation and analyzed the RNA expression of key marker genes related to pluripotency and three germ layers (ectoderm, mesoderm, and endoderm). As EB formation progressed, the expression of pluripotency markers (*Esrrb*, *Nanog*, *Pou5f1*, and *Zfp42*) decreased in both shNT and sh*Ipmk* (sh*Ipmk1* in *Figure 4*) EBs, with a marginal difference (*Figure 7A*), validating that mESCs successfully differentiated and lost self-renewal capacity. Notably, we detected significantly decreased levels of all endodermal markers (*Foxa1*, *Foxa2*, *Gata6*, and *Sox17*) upon IPMK depletion in undifferentiated EBs at day 0, and these decreased levels were maintained throughout the differentiation process (*Figure 7B*). The ectodermal marker *Fgf5* showed decreased expression upon IPMK depletion after day 4, whereas other ectodermal markers (*Otx2* and *Zic1*) remained unchanged upon IPMK depletion (*Figure 7—figure supplement 1A*). Furthermore, the majority of mesodermal markers (*Brachyury*, *Cd34*, *Gata2*, and *Runx1*) exhibited slightly decreased or unchanged expression upon IPMK depletion during EB formation (*Figure 7—figure supplement 1B*). By integrating the expression profiles of three types of marker genes, we found that IPMK depletion caused the most severe impacts on the endodermal marker genes by reducing their expression (*Figure 7—figure supplement 1C*). To understand why endodermal marker genes fail to be properly induced upon IPMK depletion during EB formation, we examined the BRG1 occupancy at their promoters in mESCs. All the reduced endodermal markers (*Foxa1*, *Foxa2*, *Gata6*, and *Sox17*) in sh*Ipmk* EBs had bivalent promoters in mESCs. Notably, all the endodermal markers (*Foxa2*, *Gata6*, and *Sox17*) except *Foxa1* exhibited decreased BRG1 occupancy at their promoters upon IPMK depletion; *Foxa2* and *Gata6* were included in Cluster2, whereas *Sox17* was included in Cluster3 (*Figure 5E*). *Foxa1* had unchanged BRG1 occupancy upon IPMK depletion. These results indicate that the *Ipmk*KD-driven reduction in BRG1 occupancy at the bivalent promoters of endodermal markers in mESCs may result in their failure to be appropriately induced in EBs. Lastly, we observed that, similar to IPMK depletion, BRG1 depletion decreased the expression of *Phactr3* (bivalent promoters) and endodermal marker genes in mESCs (*Figure 7C*), suggesting that IPMK may regulate these genes through BRG1. Collectively, these data show that IPMK may function as a regulator of the proper expression of germ layers in the early stage of mESC differentiation, especially in the development of endodermal lineages.

## Discussion

Using two unbiased screening approaches (yeast two-hybrid and APEX2 proximity labeling), we identified SMARCB1 and other core subunits of the mammalian SWI/SNF complex (BRG1, BAF155, BAF170, ARID1A, and PBRM1) as IPMK-proximal/binding targets. Notably, our binary protein interaction assays showed that IPMK can directly and independently bind to the SMARCB1, BRG1, and BAF155 proteins. Furthermore, in vivo and in vitro immunoprecipitation assays confirmed that IPMK physically interacts

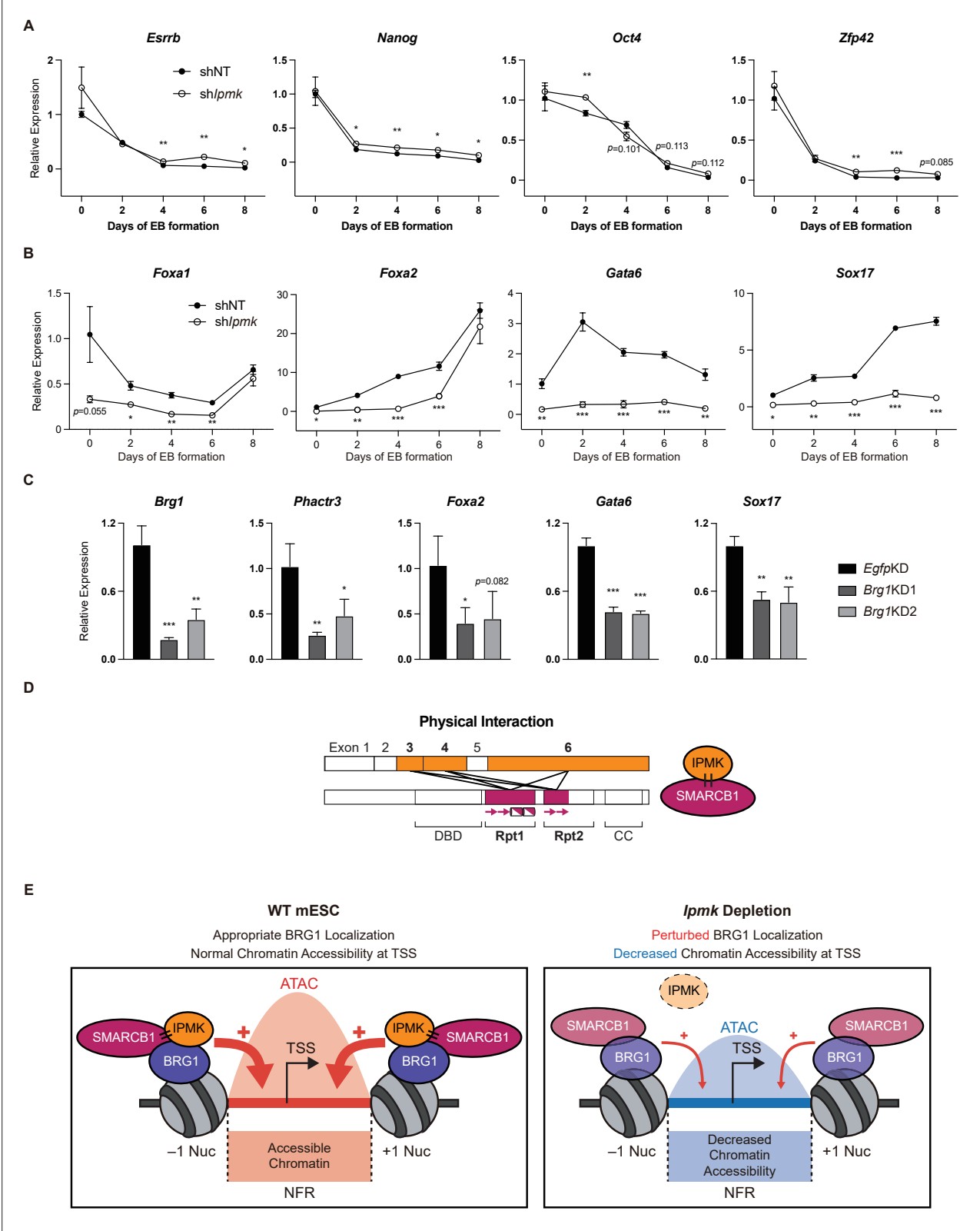

**Figure 7.** Effect of IPMK on the expression of endodermal marker genes and our proposed model depicting the function of IPMK. (**A and B**) RT-qPCR analysis of the expression of pluripotency (**A**) and endodermal (**B**) marker genes after shRNA-mediated *Ipmk* silencing at the indicated time points during embryoid body (EB) formation. Data are representative of at least three independent experiments and error bars denote the standard error of the mean obtained from three biological replicates. The expression levels were normalized to the expression of *Arbp*. (**C**) RT-qPCR analysis of *Brg1*,

*Figure 7 continued on next page*

*Figure 7 continued*

*Phactr3*, and endodermal marker gene expression after siRNA treatment (two different siRNAs against *Brg1*). Error bars denote the standard error of the mean obtained from four biological replicates. The expression levels were normalized to the expression of *β-actin*. (**A–C**) *p < 0.05, **p < 0.01, ***p < 0.001, Student's *t-test*. (**D**) A model displaying the physical interactions between IPMK and SMARCB1 (left). For these physical interactions, exons 3, 4, and 6 of IPMK (orange boxes) and the Rpt1 and Rpt2 (particularly N-terminal β sheets) domains of SMARCB1 (red boxes) are required. An additional model showing our speculation of the position of IPMK directly interacting with SMARCB1, BRG1, and BAF155 (right) within the SWI/SNF complex. (**E**) In wild-type mESCs (left), IPMK regulates appropriate BRG1 localization (probably via physical interaction with various subunits of the SWI/SNF complex) and chromatin accessibility at the nucleosome free region (NFR) of the TSS. Upon *Ipmk* depletion (right), BRG1 localization is perturbed, resulting in decreased chromatin accessibility at the NFR of TSSs.

The online version of this article includes the following source data and figure supplement(s) for figure 7:

**Source data 1.** Primary data for graphs in panel A.

**Source data 2.** Primary data for graphs in panel B.

**Source data 3.** Primary data for graphs in panel C.

**Figure supplement 1.** Effect of IPMK and BRG1 on ectodermal and mesodermal marker genes and proposed model depicting the physical interaction of IPMK and nucleosome-bound SWI/SNF complex.

**Figure supplement 1—source data 1.** Primary data for graphs in panel A.

**Figure supplement 1—source data 2.** Primary data for graphs in panel B.

**Figure supplement 1—source data 3.** Primary data for graphs in panel C.

with the SWI/SNF complex. Detailed mapping studies revealed reciprocal interactions between the Rpt domains (N-terminal β-sheets and C-terminal α-helices of Rpt1 and N-terminal β-sheets of Rpt2) of SMARCB1 and the IP kinase domain (exons 3, 4, and 6) of IPMK (***Figure 7D***). In accordance with a physical association between IPMK and the SWI/SNF complex, our CUT&RUN analysis showed that IPMK co-localizes with BRG1 globally. Surprisingly, by performing CUT&RUN and ATAC-seq, we discovered that the depletion of IPMK severely perturbed (decreased) the genome-wide BRG1 localization pattern and corresponding BRG1-mediated chromatin accessibility (ATAC-seq signals). By categorizing the promoter-TSS, we found that IPMK depletion significantly affected chromatin accessibility at bivalent promoters, which was associated with disrupted BRG1 occupancy, especially at the −1 nucleosome (Cluster2). Using RT-qPCR and mRNA-seq, we found that the IPMK loss also affects the transcription of genes exhibiting disrupted BRG1/ATAC-seq levels at their promoter-TSS. Moreover, our EB assays revealed that IPMK depletion significantly impacts endodermal gene expression during differentiation. The altered endodermal gene expression was associated with disrupted BRG1 occupancy at bivalent promoters in mESCs, suggesting that this *Ipmk*KD-driven reduction in BRG1 occupancy may cause inappropriate induction of endodermal genes in EBs. Despite our findings, the mechanism whereby IPMK modulates the localization of BRG1 (or the SWI/SNF complex) remains unclear. We think that the physical association between IPMK and the SWI/SNF complex is strongly connected with BRG1 localization, but the detailed mechanism is still elusive. IPMK may facilitate recruitment of subunits of the SWI/SNF complex from the cytoplasm to the chromatin. However, our chromatin fractionation assays indicated that IPMK depletion did not affect BRG1 or SMARCB1 occupancy in the cytoplasm or chromatin fractions (***Figure 4—figure supplement 1B and D***), excluding the above possibility. Alternatively, it is plausible that IPMK may aid the appropriate conformation of specific subsets of the SWI/SNF complex via physical binding, but further experiments are required to prove this hypothesis. Although IPMK depletion reduced the global BRG1 occupancy, our CUT&RUN results indicated that IPMK depletion has a greater impact at the CUT&RUN peaks with low enrichment of BRG1 in *Egfp* KD (WT-like) mESCs compared to highly enriched BRG1 peaks (***Figure 4B, D, G–I***). Importantly, our ATAC-seq results also corresponded to the region-specific (BRG1 Low) changes in BRG1 occupancy upon *Ipmk*KD (***Figure 4B, D, G–I***). Supporting this, IPMK depletion exhibited a greater impact on the chromatin accessibility and transcription of bivalent promoters harboring relatively low BRG1 than on H3K4me3-Only promoters, which harbor high BRG1 levels (***Figures 5 and 6***). Taken together, these results indicate that IPMK primarily regulates the BRG1 occupancy, which resembles the SWI/SNF-nucleosome interactions, and its downstream effects (i.e., chromatin accessibility and transcription) at the region where the BRG1 level is originally low in mESCs but does not affect the regions with high BRG1 levels. This contextual discrepancy in IPMK-dependent or

IPMK-independent BRG1 occupancy may provide key clues to the mechanism whereby IPMK modulates BRG1 localization, but further experiments/analyses are required.

Our study is the first to elucidate the physical association between IPMK and core subunits of the SWI/SNF complex and the first to define the molecular function of IPMK in coordinating BRG1 localization and BRG1-associated chromatin accessibility in mESCs. Based on our findings, we propose a model in which IPMK physically binds the core subunits of the SWI/SNF complex (SMARCB1, BRG1, and BAF155), maintaining appropriate BRG1 occupancy and BRG1-mediated chromatin accessibility at promoter-TSSs and thereby regulating gene expression in mESCs (*Figure 7E*). Thus, when IPMK is depleted, BRG1 localization is perturbed and chromatin accessibility decreased. Therefore, these results elucidate a critical role of IPMK in regulating BRG1 localization and BRG1-mediated chromatin accessibility through the physical association of IPMK and the core subunits of the SWI/SNF complex. We think that these novel findings will play a pivotal role in future studies on IPMK and in our understanding of the molecular mechanisms of mammalian SWI/SNF complexes, especially by providing additional clues about SWI/SNF-mediated generation of NFRs at TSSs.

# Materials and methods

## Yeast two-hybrid screening

Panbionet (Pohang, South Korea) conducted the yeast two-hybrid screening (http://panbionet.com). The full IPMK coding region (416 amino acids) was amplified by PCR. The PCR product was cloned into the pGBKT7 vector, which contains the DNA-binding domain (BD) of GAL4. *Saccharomyces cerevisiae* strain AH109 (Clontech) was co-transformed with GAL4 DNA-BD-fused IPMK and a human brain cDNA activation domain (AD) library (Clontech). Two different reporter genes (*HIS3* and *ADE2*) were used as selection markers. Yeast transformants were spread on a selection medium lacking leucine, tryptophan, and adenine or histidine (SD−LWA and SD-LWH, respectively). To confirm the interactions, the candidate prey genes were amplified by PCR or *Escherichia coli* transformation and reintroduced into the AH109 yeast strain with the IPMK bait plasmid.

## Generation of stable cell lines for APEX2-mediated proximity labeling

V5-APEX2 was PCR-amplified from the pcDNA5-Mito-V5-APEX2 plasmid, which was kindly provided by Dr. Hyun-Woo Rhee (Seoul National University). V5-APEX2 alone or IPMK-V5-APEX2 were cloned into the pcDNA5/FRT/TO plasmid (Invitrogen). Flp-In T-REx–293 (Invitrogen) cells were seeded in six-well culture plates to 70% confluency, and then co-transfected with 0.25 μg of pcDNA5/FRT/TO and 2.25 μg of pOG44 Flp recombinase expression plasmid (Invitrogen) using Lipofectamine LTX with Plus Reagent (Invitrogen). After 48 hr, the cells were transferred to 90 mm culture dishes for negative selection with 50 μg/ml Hygromycin B (Gibco) until all non-transfected cells were dead. Surviving cells were seeded at low confluency to generate cellular clones on culture plates, after which each clone was individually screened for APEX2 construct expression with or without doxycycline (Sigma Aldrich) to search for optimal cell populations with minimal uncontrolled APEX2 expression and maximal APEX2 expression under stimulation. Selected clones were then expanded and stored in liquid nitrogen for downstream experiments.

## APEX2-mediated proximity labeling

Approximately $1.4 \times 10^7$ APEX2-expressing cells were seeded in T75 culture flasks. Sixteen hours after seeding, the culture medium was exchanged for complete medium supplemented with doxycycline (100 ng/ml) for APEX2 expression. After 24 hr of induction, the cells were incubated in fresh medium containing 250 μM desthiobiotin-phenol (DBP) for 30 min in a $CO_2$ incubator at 37 °C. The cells were then moved to room temperature and hydrogen peroxide (diluted in DPBS to 1 mM; Sigma Aldrich) was added to initiate the APEX2-driven biotinylation reaction. The reaction was quenched by adding a 2 X quenching solution (20 M sodium ascorbate, 10 mM Trolox, and 20 mM sodium azide in DPBS). The cells were washed with 1 X quencher solution three times, collected by centrifugation, snap-frozen, and stored at –80 °C until lysis. DBP was synthesized as described in a previous report (*Lee et al., 2017*).

## Preparation of DBP-labeled peptides for LC-MS/MS

DBP-labeled peptides were prepared from frozen cell pellets as described previously (*Kwak et al., 2020*). Briefly, cells were lysed in lysis buffer (2% SDS, 1 X protease inhibitor cocktail [Roche], and

1 mM sodium azide in 1 X TBS) and excess DBP was eliminated through repeated acetone precipitation. The resulting protein precipitates were solubilized in 50 mM ammonium bicarbonate and quantified. A total of 4 mg of cellular protein was then denatured, reduced, alkylated, and digested by trypsin into peptides. The tryptic DBP-labeled peptides were bound to streptavidin beads (Pierce) and collected with elution buffer (80% acetonitrile, 0.2% trifluoroacetic acid, 0.1% formic acid in MS-grade water). Solvents were completely evaporated on a SpeedVac for 3 hr, and the resulting peptides were stored at –20 °C until required for LC-MS/MS analysis.

## LC-MS/MS

The tryptic peptides were analyzed by LC-MS/MS. All mass analyses were performed on a Q Exactive Plus orbitrap mass spectrometer (Thermo Fisher Scientific) equipped with a nanoelectrospray ion source. To separate the peptide mixture, we used a C18 reverse-phase HPLC column (500 mm ×75 μm ID) using an acetonitrile/0.1% formic acid gradient from 3.2% to 26% for 120 min at a flow rate of 300 nl/min. For MS/MS analysis, the precursor ion scan MS spectra (m/z 400~2000) were acquired in the Orbitrap at a resolution of 70,000 at m/z 400 with an internal lock mass. The 15 most intensive ions were isolated and fragmented by high-energy collision-induced dissociation (HCD).

## LC-MS/MS data processing

All MS/MS samples were analyzed using the Sequest Sorcerer platform (Sage-N Research, San Jose, CA), which was set up to search the *Homo sapiens* protein sequence database [20,675 entries, UniProt (http://www.uniprot.org/)]. This database includes frequently observed contaminants assuming the digestion enzyme is trypsin. Sequest was searched with a fragment ion mass tolerance of 1.00 Da and a parent ion tolerance of 10.0 ppm. Carbamidomethylation of cysteine was specified as a fixed modification. Oxidation of methionine and acetylation of the N-terminus, biotinylation of lysine, and DBP of tyrosine were specified in Sequest as variable modifications. Scaffold (Version 4.11.0, Proteome Software Inc, Portland, OR) was used to validate the MS/MS-based peptide and protein identifications. Peptide identifications were accepted if they could be established at > 93.0% probability to achieve a false discovery rate (FDR) < 1.0% by the Scaffold Local FDR algorithm. Protein identifications were accepted if they could be established at > 92.0% probability to achieve an FDR < 1.0% and contained at least two identified peptides. Protein probabilities were assigned by the Protein Prophet algorithm (*Nesvizhskii et al., 2003*). Proteins that contained similar peptides and could not be differentiated based on MS/MS analysis alone were grouped to satisfy the principles of parsimony. Proteins were annotated with GO terms from NCBI (downloaded November 23, 2019) (*Ashburner et al., 2000*).

## Plasmids

The cDNAs for human IPMK (NCBI Gene ID 253430) and human SMARCB1 (NCBI Gene ID 6598) were obtained from Open Biosystems and Bioneer (Daejeon, South Korea), respectively. IPMK and SMARCB1 cDNA constructs were amplified by PCR and the products were cloned into pCMV-GST and pcDNA3.1-FLAG vectors. Every construct was confirmed by DNA sequencing.

## In vitro binding assay

Recombinant human IPMK was purified as described previously (*Lee et al., 2020*). Briefly, human IPMK was expressed in Sf9 insect cells using a baculovirus system and harvested with lysis buffer (300 mM NaCl, 50 mM Tris [pH 8.0], 5% glycerol, and 1 mM phenylmethylsulfonylfluoride [PMSF]). The freeze-thaw lysis method with liquid nitrogen was applied to the cells and the supernatants were collected after centrifugation at 18,000 rpm for 90 min. Ni-NTA agarose (Qiagen) was applied with 20 mM imidazole and incubated for 2 hr. The protein was eluted with 100 mM imidazole and the N-terminal HIS-tag was removed with TEV protease, followed by further purification with HiTrap and Superdex columns (GE Healthcare). Human SMARCB1 was translated in vitro using the TNT Quick Coupled Transcription/Translation System (L1170, Promega). pcDNA3.1-FLAG-SMARCB1 (1 μg) was incubated at 30 °C for 90 min with 20 μM methionine and TNT T7 Quick Master Mix. Translated FLAG-SMARCB1 was incubated with anti-FLAG M2 affinity gel (A2220, Sigma Aldrich), and then IPMK protein was added and incubated with rotation at 4 °C.

## Recombinant IPMK protein purification

For GST-tagged protein, human IPMK cDNA was subcloned into pGEX4T plasmid (Sigma Aldrich), expressed in *E. coli*, and purified on Glutathione Sepharose 4B beads (GE Healthcare) as described

previously (*Kim and Roeder, 2011*). For FLAG-tagged proteins, IPMK cDNAs were subcloned into pFASTBAC1 plasmid (Thermo Fisher Scientific) with an N-terminal FLAG epitope and baculoviruses were generated according to the manufacturer's instructions. Proteins were expressed in Sf9 insect cells and purified on M2 agarose (Sigma Aldrich) as described previously (*Kim and Roeder, 2011*).

## BAF complex purification

The FLAG-DPF2 cell line was selected from HEK293T cells transfected with a FLAG-DPF2-pCAG-IP plasmid. Derived nuclear extracts (*Dignam et al., 1983*) were incubated with M2 agarose in binding buffer (20 mM Tris-HCl [pH 7.3], 300 mM KCl, 0.2 mM EDTA, 25% glycerol, 1.5 mM MgCl$_2$, 10 mM 2-mercaptoethanol. and 0.2 mM PMSF) at 4 °C for 4 hr. After extensive washing with wash buffer (20 mM Tris-HCl [pH 7.9], 150 mM NaCl, 0.2 mM EDTA, 5% glycerol, 2 mM MgCl$_2$, 10 mM 2-mercaptoethanol, 0.2 mM PMSF, and 0.1% NP-40), complexes were eluted with wash buffer containing 0.25 mg/ml FLAG peptide. Eluted complexes were fractionated by a 10%–30% glycerol gradient and the fractions containing intact BAF complex were combined and concentrated using Amicon Ultra-4 centrifugal filter (Millipore).

## Protein interaction assays

For GST pull-down assays, 2 µg of GST or GST-tagged IPMK immobilized on Glutathione Sepharose 4B beads was incubated with 200 ng of purified BAF complex in binding buffer (20 mM Tris-HCl [pH 7.9], 150 mM KCl, 0.2 mM EDTA, 20% glycerol, 0.05% NP-40, and 0.2 mg/ml BSA) at 4 °C for 3 hr. Beads were extensively washed with binding buffer without BSA and bound proteins were analyzed by immunoblotting. For binary protein interaction assays following baculovirus-mediated expression, Sf9 cells were infected with baculoviruses expressing FLAG-IPMK and untagged BAF complex subunit. After 2 days, total cell extracts were prepared by sonication in lysis buffer (20 mM Tris-HCl [pH 7.9], 300 mM NaCl, 0.2 mM EDTA, 15% glycerol, 2 mM MgCl$_2$, 1 mM DTT, 1 mM PMSF, and protease inhibitor cocktail [Roche]). Following clarification by centrifugation, cell extracts were incubated with M2 agarose at 4 °C for 3 hr and, after extensive washing with wash buffer (20 mM Tris-HCl [pH 7.9], 150 mM NaCl, 0.2 mM EDTA, 15% glycerol, 2 mM MgCl$_2$, 1 mM DTT, 1 mM PMSF, and 0.1% NP-40), bound proteins were analyzed by immunoblotting.

## Immunoblotting, immunoprecipitation, and GST pull-down

For immunoblot analyses, cells were washed twice with PBS and lysed in lysis buffer (1% Triton X-100, 120 mM NaCl, 40 mM Tris-HCl [pH 7.4], 1.5 mM sodium orthovanadate, 50 mM sodium fluoride, 10 mM sodium pyrophosphate, 1 mM EDTA, and protease inhibitor cocktail [Roche]). Cell lysates were incubated at 4 °C for 10 min and the supernatants were collected by centrifugation at 13,000 rpm for 10 min. Protein concentrations were determined by the Bradford protein assay (Bio-Rad) or bicinchoninic acid (BCA) assay (Thermo Fisher Scientific). Protein lysates (20 µg) were separated by size, transferred to nitrocellulose membranes, and blotted with primary antibodies and secondary antibodies. The horseradish peroxidase (HRP) signals were visualized using the Clarity ECL substrate (Bio-Rad) and SuperSignal West Femto Maximum Sensitivity Substrate (Thermo Fisher Scientific) and measured by a ChemiDoc imaging system (Bio-Rad). For immunoprecipitation, 2 mg of total protein was incubated with 5 µg of primary antibodies for 16 hr with rotation at 4 °C. TrueBlot beads (10 µl; Rockland Immunochemicals) were added and incubated for an additional hour. The samples were washed three times with lysis buffer and prepared for immunoblotting. For GST pull-down assays, 10 µl of glutathione agarose beads (Incospharm) were added to 2 mg of total cell lysate and incubated for 16 hr with rotation at 4 °C. The samples were then washed three times with lysis buffer and prepared for immunoblotting.

## Cell culture and cell line production

E14Tg2a mESCs were maintained under feeder-free conditions. Briefly, the cells were cultured on gelatin-coated cell culture dishes in an mESC culture medium consisting of Glasgow's minimum essential medium (GMEM) containing 10% knockout serum replacement, 1% non-essential amino acids, 1% sodium pyruvate, 0.1 mM β-mercaptoethanol (all from Gibco), 1% FBS, 0.5% antibiotic-antimycotic (both from Hyclone), and 1,000 units/ml LIF (ESG1106, Millipore). The mESCs were maintained at 37 °C in humidified air with 5% CO$_2$. NIH3T3 cells, MEFs, and HEK293T cells were grown in high-glucose

DMEM supplemented with 10% FBS, 2 mM L-glutamine, and penicillin/streptomycin (100 mg/ml), and maintained at 37 °C in a humid atmosphere with 5% $CO_2$. To generate tamoxifen-inducible IPMK knockout mice, $Ipmk^{fl/fl}$ mice were mated with UBC-Cre-ERT2 mice (The Jackson Laboratory). The MEFs were immortalized by transfection with an SV40 large T-antigen plasmid, and IPMK depletion was achieved by adding 1 μM 4-hydroxytamoxifen for 48 hr. FLAG epitope-tagged mESCs were generated as described previously (*Savic et al., 2015*). Briefly, the 3xFlag-P2A-Puromycin epitope tagging donor construct (pFETCh-Donor), CRISPR guide RNAs (gRNAs), and Cas9-expressing plasmids were manufactured by ToolGen (Seoul, Korea). The mESCs were transfected using FUGENE HD (E2311, Promega), selected using puromycin (A11138-03, Gibco), and expanded. MEFs were transfected with the donor construct containing the neomycin resistance gene using Turbofect (R0533, Thermo Fisher Scientific) and selected using G418 (11811023, Gibco). The pLKO.1-TRC cloning vector (Addgene) was used as a backbone for the expression of genes of interest. The shRNA targeting *Ipmk* was purchased from Sigma Aldrich. Lentiviruses were generated in HEK293T cells by transfecting the lentiviral plasmids together with packaging plasmid (psPAX2) and envelope-expressing plasmid (pMD2.G). Lentiviruses were added to mESCs after 48 hr and incubated for 8 hr. After 40 hr, mESCs with stable gene expression were selected by growing cells in mESC medium supplemented with Hygromycin B (Invitrogen).

## RNA interference

Control siRNA (scRNA) and siRNAs against *Egfp* (sense: 5'-GUUCAGCGUGUCCGGCGAG-3', antisense: 5'-CUCGCCGGACACGCUGAAC-3'), *Ipmk* (sense: 5'-CAGAGAGGUCCUAGUUAAUUUCA-3', antisense: 5'-AGUGAAAUUAACUAGGACCUCUCUGUU-3'), and *Brg1* (*Brg1*KD1 sense: 5'- GUCA GACAGUAAUAAAUUAAAGCAA-3', antisense: 5'- UUGCUUUAAUUUAUUACUGUCUGAC-3'; *Brg1*KD2 sense: 5'- CCGUGCAACGAACCAUAAA-3', antisense: 5'- UUUAUGGUUCGUUGCAC-GG-3') were synthesized and annealed by Bioneer (Daejeon, Korea), and siRNA against *Smarcb1* was purchased from Sigma Aldrich. The mESCs and MEFs were transfected with 50 nM of the corresponding siRNA using DharmaFECT I (T-2001–03, Dharmacon) according to the manufacturer's instructions. Briefly, the cells were seeded in six-well plates. One day later, 50 nM of siRNA and DharmaFECT reagent were diluted in Opti-MEM (Gibco), incubated separately at 25 °C for 5 min, and then mixed together. The mixtures were incubated at 25 °C for 20 min and added to the cell cultures. The culture medium was replaced after 24 hr and the transfected cells were harvested at 48 hr after transfection.

## Chromatin fractionation

Chromatin acid extraction was performed as described previously (*Zhong et al., 2013*). The mESCs or MEFs were collected and washed with PBS and resuspended with lysis buffer (10 mM HEPES [pH 7.4], 10 mM KCl, 0.05% NP-40, 1 mM sodium orthovanadate, and protease inhibitor cocktail [Roche]). The cell lysates were incubated for 20 min on ice and centrifuged at 13,000 rpm. The supernatant contained the cytoplasmic proteins, and the pellet with nuclei was washed once with lysis buffer and centrifuged at 13,000 rpm for 10 min. The nuclei were resuspended in low-salt buffer (10 mM Tris-HCl [pH 7.4], 0.2 mM $MgCl_2$, 1% Triton-X 100, 1 mM sodium orthovanadate, and protease inhibitor cocktail) and incubated on ice for 15 min. After 10 min of centrifugation, the supernatant contained the nucleoplasmic proteins and the pellet contained the chromatin. The chromatin was then resuspended in 0.2 N HCl for 20 min on ice, centrifuged at 13,000 rpm for 10 min, and neutralized with 1 M Tris-HCl (pH 8.0). Protein concentrations were determined and the proteins were subjected to immunoblotting.

## CUT&RUN

CUT&RUN assays were performed as described previously (*Meers et al., 2019*; *Skene and Henikoff, 2017*), with minor modifications. Briefly, 4 million mESCs were harvested and washed three times with 1.5 ml wash buffer (20 mM HEPES [pH 7.5], 150 mM NaCl, 0.5 mM Spermidine). Cells were bound to activated concanavalin A-coated magnetic beads at 25 °C for 10 min on a nutator, and then permeabilized with antibody buffer (wash buffer containing 0.05% digitonin and 4 mM EDTA). The bead-cell slurry was incubated with 3 μl of relevant antibody (see below) in a 150 μl volume at 25 °C for 2 hr on a nutator. After two washes in 1 ml Dig-wash buffer (wash buffer containing 0.05% digitonin), the beads were resuspended in 150 μl pAG/MNase and incubated at 4 °C for 1 hr on a nutator. After two washes in 1 ml Dig-wash buffer, the beads were gently vortexed with 100 μl Dig-wash buffer.

Tubes were chilled to 0 °C for 5 min and ice-cold 2.2 mM CaCl$_2$ was added while gently vortexing. Tubes were immediately placed on ice and incubated at 4 °C for 1 hr on a nutator, followed by the addition of 100 µl 2xSTOP buffer (340 mM NaCl, 20 mM EDTA, 4 mM EGTA, 0.05% digitonin, 0.1 mg/ml RNase A, 50 µg/ml glycogen) and incubated at 37 °C for 30 min on a nutator. Beads were placed on a magnet stand, the liquid was removed to a fresh tube, and 2 µl 10% SDS and 2.5 µl proteinase K (20 mg/ml) were added before incubation at 50 °C for 1 hr. DNA was extracted using phenol chloroform as described at https://www.protocols.io/view/cut-amp-run-targeted-in-situ-genome-wide-profiling-zcpf2vn. CUT&RUN libraries were prepared using a NEXTflex ChIP-seq Library kit (5143–02, Bioo Scientific) according to the manufacturer's guidelines. The libraries were then sequenced using an Illumina Novaseq 6000 platform. Libraries were generated from two sets of biological replicates.

## Antibodies

Antibodies against FLAG (F1804, Sigma Aldrich), IPMK (custom rabbit polyclonal antibody, raised against a mouse IPMK peptide corresponding to amino acids 295–311 [SKAYSTHTKLYAKKHQS; Covance]; *Kim et al., 2011*), SMARCB1 (A301-087, Bethyl), BRG1 (ab110641, Abcam), BAF155 (11956, Cell Signaling Technology), BAF170 (12760, Cell Signaling Technology), BAF250A (12354, Cell Signaling Technology), BRM (11966, Cell Signaling Technology), PBAF/PBRM (A301-591A, Bethyl), a-TUBULIN (T5169, Sigma Aldrich), GST (2622, Cell Signaling Technology), LaminB1 (sc-365214, Santa Cruz Biotech), histone H3 (05–499, Sigma Aldrich), GAPDH (sc-32233, Santa Cruz Biotech), anti-DPF2 (ab128149, Abcam), anti-SMARCE1 (ab137081, Abcam), anti-SS18L1 (ab227535, Abcam), anti-ACTL6A (sc-137062, Santa Cruz Biotech), anti-SMARCD1 (sc-135843, Santa Cruz Biotech), anti-BCL7A (HPA019762, Atlas Antibodies), and anti-ACTB (TA811000, Origene) were used for immunoblotting. Antibodies against IPMK (homemade), SMARCB1 (A301-087, Bethyl), and rabbit IgG isotype control (02–6102, Invitrogen) and anti-FLAG M2 affinity gel (A2220, Sigma Aldrich) were used for immunoprecipitation, and GST (2622, Cell Signaling Technology) was used for pull-down assays. Antibodies against FLAG (F7425, Sigma Aldrich), BRG1 (ab110641, Abcam), and IgG (homemade) were used for CUT&RUN assays.

## ATAC-seq

ATAC-seq libraries were prepared as described previously (*Buenrostro et al., 2013*; *Buenrostro et al., 2015*), with minor modifications. Briefly, 50,000 mESCs were harvested, washed with cold PBS, lysed with cold lysis buffer, and immediately centrifuged. The nuclear pellets were resuspended in 25 µl of 2 X tagmentation reaction buffer (10 mM Tris [pH 8.0], 5 mM MgCl$_2$, 10% dimethylformamide), 23 µl of nuclease-free water, and 2 µl of Tn5 transposase (generated in-house), and incubated at 37 °C for 30 min. The samples were immediately purified using a QIAquick PCR purification kit (28106, Qiagen). The libraries were pre-enriched for five cycles using the KAPA HiFi Hotstart ready mix (KK2601, Kapa Biosystems), and the threshold cycle (Ct) was monitored using qPCR to determine the additional enrichment cycles, which were then applied. The final libraries were purified again using a QIAquick PCR purification kit and sequenced using an Illumina Novaseq 6,000 platform. The libraries were generated from two sets of biological replicates.

## H3K4me3-Only and bivalent promoter-TSSs

The list of 47,382 mouse genes was obtained from the UCSC Genome Browser [Table browser, mm10, group: Genes and Gene Prediction, track: NCBI RefSeq, table: UCSC RefSeq (refGene), region: genome]. Among these genes, we selected protein-coding genes (gene name starting with NM_) that are longer than 2 kb. To classify the promoter-TSS regions precisely, we removed redundancies by merging the genes with the exact same TSSs into the same group. By doing this, we obtained 23,927 mouse promoter-TSS regions. To categorize the promoter-TSS regions according to their histone modifications status, we calculated the H3K4me3 (accession number: GSM254000 and GSM254001) and H3K27me3 (accession number: GSM254004 and GSM254005) ChIP-seq intensity in a −500/ + 1000 bp window around the promoter-TSS regions. The −500/ + 1000 bp window range was also applied in a previous study (*de Dieuleveult et al., 2016*). Next, we divided the promoter-TSSs into two groups based on the first quartile (Q1) value of the H3K4me3 ChIP-seq intensity in the whole promoter-TSS regions: H3K4me3-Low and H3K4me3-High. Next, we divided the H3K4me3-High promoter-TSSs into two groups based on the third quartile (Q3) value of the H3K27me3 ChIP-seq

intensity in the whole promoter-TSS regions: H3K4me3-Only and bivalent. Thus, we categorized three types of promoter-TSSs: 5,982 H3K4me3-Low, 12,305 H3K4me3-Only (high H3K4me3 and low H3K27me3), and 5,640 bivalent (high H3K4me3 and high H3K27me3). We obtained similar results when using the publicly released H3K4me3 and H3K27me3 ChIP-seq data (ENCODE, Ross Hardison, ENCSR212KGS and ENCSR059MBO). We confirmed that the percentages of three promoter-TSS types are similar to the results obtained in a previous study (*de Dieuleveult et al., 2016*).

## mRNA purification, mRNA-seq, and rt-qPCR

Total RNA was purified from mESCs using TRIzol reagent (Invitrogen) according to the manufacturer's instructions. Briefly, mESCs cultured in six-well plates were harvested and homogenized with 1 ml of TRIzol reagent. Chloroform (200 µl/sample) was added and the samples vigorously mixed by hand for 15 s and incubated at 25 °C for 2 min. The mixtures were centrifuged at 12,000 rpm for 15 min at 4 °C, and 500 µl of each aqueous phase was transferred to a new Eppendorf tube and mixed with equal volumes of isopropanol. The mixtures were incubated at 25 °C for 10 min to precipitate the total RNA samples. The samples were then centrifuged at 12,000 rpm for 10 min at 4 °C, washed with 75% ethanol, and centrifuged again at 10,000 rpm for 5 min at 4 °C. The RNA pellets were dried and dissolved in RNase-free water. For mRNA-seq library preparation, mRNA was isolated from total RNA using a Magnetic mRNA Isolation Kit (S1550S, NEB), and libraries were prepared using a NEXTflex Rapid Directional RNA-seq Kit (5138–08, Bioo Scientific). The libraries were sequenced using an Illumina HiSeq 2,500 system. The libraries were generated from two sets of biological replicates. First-strand complementary DNA was synthesized from total RNA using reverse transcriptase (Enzynomics). RT-qPCR analyses was performed using SYBR Green Master Mix (Toyobo) and the StepOnePlus Real-Time PCR System (Applied Biosystems). The expression levels of genes of interest were normalized to those of a housekeeping gene and are presented as fold changes over baseline using the $\Delta\Delta C_t$ method.

## Embryoid body formation

To induce EB formation, E14Tg2a cells were detached, pelleted by centrifugation, and resuspended in mESC medium in the absence of LIF and β-mercaptoethanol. Cell drops (2500 cells/25 µl) were hanging cultured on the lids of the dishes. After 2 days, the drops were collected and transferred to gelatin-coated dishes. EBs were maintained in mESC medium without LIF and β-mercaptoethanol for 8 days. The medium was changed every 2 days and aggregate growth was monitored under a microscope. The resulting EBs were harvested at the indicated time points.

## Data processing and analysis

For CUT&RUN analysis, raw reads were aligned to the mouse genome (mm10) using Bowtie2 (version 2.2.9) (*Langmead and Salzberg, 2012*) with the following parameter according to the previous study (GSM2247138): `--trim3 125 --local --very-sensitive-local --no-unal --no-mixed --no-discordant -q --phred33 -I 10 X 700`. For ATAC-seq analysis, raw reads were aligned to the mouse genome (mm10) using Bowtie2 (version 2.2.9) with the following parameter: --very-sensitive -X 100 –local. For mRNA-seq analysis, raw reads were aligned to the mouse genome (mm10) using STAR (version 2.5.2 a) with default parameters (*Dobin et al., 2013*). Generally, we used MACS2 (*Zhang et al., 2008*) to convert the aligned BAM files into bedGraph files and normalized the data with respect to the total read counts. We used bedGraphToBigWig (*Kent et al., 2010*) to convert the bedGraph files into bigWig files. The bigWig files were used as input files for bwtool (matrix and aggregate option) to quantify the intensity (e.g. heatmaps or average line plots) of the relevant sequencing data (*Pohl and Beato, 2014*). All of our raw data (fastq files) were confirmed to be of good quality using FastQC (http://www.bioinformatics.babraham.ac.uk/projects/fastqc/). For our CUT&RUN analysis, we used MACS2 (callpeak option, p < 0.005) to identify the peaks (or binding sites) of proteins of interest using IgG as background. The CUT&RUN data were also subjected to HOMER annotatPeaks.pl (*Heinz et al., 2010*) to elucidate the genomic content within BRG1/IPMK-binding sites. For our mRNA-seq analyses, we used Cufflinks (Cuffdiff option, fr-firststrand) to assess the expression levels and identify DEGs (*Trapnell et al., 2010*). Box plots, volcano plots, and other plots were drawn with R (ggplot2) (*Wickham, 2009*), and heatmaps were drawn with Java TreeView (*Saldanha, 2004*).

The examples of our genome-wide data were visualized using the Integrative Genomics Viewer (IGV) (*Robinson et al., 2011*).

## Public data acquisition

Publicly released ChIP-seq data were downloaded from the NCBI GEO DataSets database as sra or fastq files. The sra files were converted to fastq files using the SRA Toolkit (https://trace.ncbi.nlm.nih.gov/Traces/sra/sra.cgi?view=software). Both the public datasets and our data were then analyzed using the same methods.

# Acknowledgements

We thank members of the S.K. and D.L. laboratory for their helpful discussions.

# Additional information

### Competing interests

Jiyoon Beon: The authors have no potential conflicts of interest to disclose. The other authors declare that no competing interests exist.

### Funding

| Funder | Grant reference number | Author |
| --- | --- | --- |
| National Research Foundation of Korea | NRF-2020R1A2C3005765 | Seyun Kim |
| National Research Foundation of Korea | NRF-2018R1A5A1024261 | Seyun Kim Daeyoup Lee |

The funders had no role in study design, data collection and interpretation, or the decision to submit the work for publication.

### Author contributions

Jiyoon Beon, Sungwook Han, Conceptualization, Data curation, Formal analysis, Investigation, Validation, Visualization, Writing - original draft, Writing - review and editing; Hyeokjun Yang, Formal analysis, Investigation, Methodology; Seung Eun Park, Formal analysis, Investigation, Methodology, Performed and analyzed APEX2-proximity labeling; Kwangbeom Hyun, Formal analysis, Investigation, Methodology, Performed experiments for figure 2B and Figure 2-figure supplement 1H and I; Song-Yi Lee, Hyun-Woo Rhee, Methodology; Jeong Kon Seo, Investigation, Methodology, Performed mass spectrometry; Jaehoon Kim, Methodology, Resources; Seyun Kim, Conceptualization, Funding acquisition, Project administration; Daeyoup Lee, Conceptualization, Funding acquisition, Project administration, Supervision, Writing - review and editing

### Author ORCIDs

Jiyoon Beon http://orcid.org/0000-0002-6347-9122
Sungwook Han http://orcid.org/0000-0003-3944-3266
Seyun Kim http://orcid.org/0000-0003-0110-9414
Daeyoup Lee http://orcid.org/0000-0003-2006-1823

### Decision letter and Author response

Decision letter https://doi.org/10.7554/eLife.73523.sa1
Author response https://doi.org/10.7554/eLife.73523.sa2

# Additional files

### Supplementary files

• Supplementary file 1. Yeast two-hybrid screening assay using IPMK as bait.
• Supplementary file 2. Enriched protein complex-based sets from APEX2-mediated proximity

labeling.
- Supplementary file 3. Appendix_ Sequence-based reagents.
- Transparent reporting form

## Data availability

Sequencing data have been deposited in GEO under accession codes GSE158525.

The following dataset was generated:

| Author(s) | Year | Dataset title | Dataset URL | Database and Identifier |
|---|---|---|---|---|
| Han S, Beon J, Park SE, Hyun K, Lee S, Rhee H, Seo JK, Kim J, Kim S, Lee D | 2020 | IPMK physically binds to the SWI/SNF complex and modulates BRG1 occupancy | https://www.ncbi.nlm.nih.gov/geo/query/acc.cgi?acc=GSE158525 | NCBI Gene Expression Omnibus, GSE158525 |

The following previously published datasets were used:

| Author(s) | Year | Dataset title | Dataset URL | Database and Identifier |
|---|---|---|---|---|
| Han S, Lee H, Lee AJ, Kim S, Jung I, Koh GY, Kim T, Lee D | 2021 | Chd4 regulates intra-chromatin looping by concealing CTCF sites from B2 SINEs in mESCs | https://www.ncbi.nlm.nih.gov/geo/query/acc.cgi?acc=GSE172392 | NCBI Gene Expression Omnibus, GSE172392 |
| Hmitou I, de Dieuleveult M, Depaux A, Chantalat S, Yen K, Pugh BF, Gérard M | 2015 | Genome-wide distribution and function of ATP-dependent chromatin remodelers in embryonic stem cells | https://www.ncbi.nlm.nih.gov/geo/query/acc.cgi?acc=GSE64825 | NCBI Gene Expression Omnibus, GSE64825 |

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

# Appendix 1

## Appendix 1—key resources table

| Reagent type (species) or resource | Designation | Source or reference | Identifiers | Additional information |
|---|---|---|---|---|
| Gene (*Homo-sapiens*) | IPMK | Open Biosystems | | NCBI Gene ID 253430 |
| Gene (*Homo-sapiens*) | SMARCB1 | Bioneer | | NCBI Gene ID 6598 |
| Strain, strain background (*Escherichia coli*) | DH5α | Enzynomics | Cat# CP010 | |
| Cell line (*Saccharomyces cerevisiae*) | AH109 | Clontech | | See Materials and methods. Cell line maintained in Panbionet. |
| Cell line (*M. musculus*) | E14Tg2a | ATCC | Cat# CRL-1821 | |
| Cell line (*M. musculus*) | MEF | This paper | | See Materials and methods. Cell line maintained in S. Kim lab. |
| Cell line (*M. musculus*) | NIH3T3 | ATCC | Cat# CRL-1658 | |
| Cell line (*Homo-sapiens*) | HEK293T | ATCC | Cat# CRL-3216 | |
| Cell line (*Homo-sapiens*) | Flp-In T-REx–293 | Invitrogen | Cat# R78007 | |
| Cell line (*Spodoptera frgiperda*) | Sf9 | Invitrogen | Cat# 11496015 | |
| Transfected construct (*M. musculus*) | siRNA to *Egfp* | Bioneer | | transfected construct (mouse) |
| Transfected construct (*M. musculus*) | siRNA to *Ipmk* | Bioneer | | transfected construct (mouse) |
| Transfected construct (*M. musculus*) | siRNA to *Smarcb1* | Dharmacon/Thermo Fisher Scientific | Cat# 4390771 | transfected construct (mouse) |
| Transfected construct (*M. musculus*) | siRNA to *Brg1* | Bioneer | | transfected construct (mouse) |
| Antibody | anti-FLAG (Mouse monoclonal) | Sigma Aldrich | Cat# F1804 | WB (1:1000) |
| Antibody | anti-SMARCB1 (Rabbit polyclonal) | Bethyl | Cat# A301-087 | WB (1:1000) IP (3 ug) |
| Antibody | anti-BRG1 (Rabbit monoclonal) | Abcam | Cat# Ab110641 | WB (1:1000) |
| Antibody | anti-BAF155 (Rabbit monoclonal) | Cell Signaling Technology | Cat# 11,956 | WB (1:1000) |
| Antibody | anti-BAF170 (Rabbit monoclonal) | Cell Signaling Technology | Cat# 12,760 | WB (1:1000) |
| Antibody | anti-BAF250A (Rabbit monoclonal) | Cell Signaling Technology | Cat# 12,354 | WB (1:1000) |
| Antibody | anti-PBAF/PBRM (Rabbit polyclonal) | Bethyl | Cat# A301-591A | WB (1:1000) |
| Antibody | anti-α-Tubulin (Mouse monoclonal) | Sigma Aldrich | Cat# T5169 | WB (1:3000) |
| Antibody | anti-IPMK (Rabbit polyclonal) | custom rabbit polyclonal | DOI: 10.1126/ sciadv.1602296 | WB (1:1000) IP (3 ug) |
| Antibody | anti-GST (Rabbit polyclonal) | Cell Signaling Technology | Cat# 2,622 | WB (1:1000) |
| Antibody | anti-Histone H3 (mouse monoclonal) | Sigma Aldrich | 05–499 | WB (1:3000) |
| Antibody | anti-GAPDH (mouse monoclonal) | Santa Cruz Biotech | Cat# sc-32233 | WB (1:3000) |
| Antibody | anti-LaminB1 (mouse monoclonal) | Santa Cruz Biotech | Cat# sc-365214 | WB (1:1000) |

*Appendix 1 Continued on next page*

*Appendix 1 Continued*

| Reagent type (species) or resource | Designation | Source or reference | Identifiers | Additional information |
|---|---|---|---|---|
| Antibody | anti-DPF2 (rabbit polyclonal) | Abcam | Cat# ab128149 | WB (1:1000) |
| Antibody | anti-SMARCE1 (rabbit monoclonal) | Abcam | Cat# ab137081 | WB (1:1000) |
| Antibody | anti-SS18L1 (rabbit polyclonal) | Abcam | Cat# ab227535 | WB (1:1000) |
| Antibody | anti-ACTL6a (mouse monoclonal) | Santa Cruz Biotech | Cat# sc-137062 | WB (1:1000) |
| Antibody | anti-SMARCD1 (mouse monoclonal) | Santa Cruz Biotech | Cat# sc-135843 | WB (1:1000) |
| Antibody | anti-BCL7A (rabbit polyclonal) | Atlas Antibodies | Cat# HPA019762 | WB (1:1000) |
| Antibody | anti-ACTB (mouse monoclonal) | Origene | Cat# TA811000 | WB (1:1000) |
| Antibody | rabbit IgG isotype control (rabbit isotype control) | Invitrogen | 02–6102 | IP (3 ug) |
| Recombinant DNA reagent | pGBKT7-GAL4-DNA-BD-fused IPMK (plasmid) | This paper | | See Materials and methods. IPMK cloned into BD-containing pGBKT7 vector. |
| Recombinant DNA reagent | human brain cDNA activation domain (AD) library | Clontech | Cat# 630,486 | |
| Recombinant DNA reagent | pLKO.1-hygro (plasmid) | Addgene | Cat# 24,150 | |
| Recombinant DNA reagent | pcDNA5-Mito -V5-APEX2 (plasmid) | This paper | | See Materials and methods. Kindly provided by H. Rhee lab. |
| Recombinant DNA reagent | pcDNA5/FRT/TO vector (plasmid) | Invitrogen | Cat# V652020 | |
| Recombinant DNA reagent | pOG44 Flp recombinase expression vector (plasmid) | Invitrogen | Cat# V600520 | |
| Recombinant DNA reagent | pCMV-GST (plasmid) | This paper | DOI: 10.1126/sciadv.1602296 | See Materials and Methods. |
| Recombinant DNA reagent | pcDNA3.1-FLAG (plasmid) | This paper | DOI: 10.1126/sciadv.1602296 | See Materials and methods |
| Recombinant DNA reagent | pGEX4T (plasmid) | Sigma Aldrich | Cat# GE28-9545-52 | |
| Recombinant DNA reagent | pFASTBAC1 (plasmid) | Thermo Fisher Scientific | Cat# 10359016 | |
| Recombinant DNA reagent | FLAG-DPF2-pCAG-IP (plasmid) | This paper | | See Materials and methods. Kindly provided by J. Kim lab. |
| Sequence-based reagent | *Ipmk* shRNA #1 | Sigma Aldrich | TRC Clone ID TRCN0000360808 | . |
| Sequence-based reagent | *Ipmk* shRNA #2 | Sigma Aldrich | TRC Clone ID TRCN0000360733 | |
| Sequence-based reagent | shNT_F | This paper | Oligo sequence used for cloning pLKO.1 shNT (negative control) | CCGGTCCTAAG GTTAAGTCGCCCTCG CTCGAGCGAG GGCGACTTAA CCTTAGGTTTTTG |
| Sequence-based reagent | shNT_R | This paper | Oligo sequence used for cloning pLKO.1 shNT (negative control) | AATTCAAAAACCT AAGGTTAAGTCGC CCTCGCTCGAG CGAGGGCG ACTTAACCTTAGGA |
| Peptide, recombinant protein | Streptavidin | Thermo Fisher | Cat# 434,302 | |

*Appendix 1 Continued on next page*

*Appendix 1 Continued*

| Reagent type (species) or resource | Designation | Source or reference | Identifiers | Additional information |
|---|---|---|---|---|
| Commercial assay or kit | Bradford protein assay | Bio-rad | Cat# 5000006 | |
| Commercial assay or kit | BCA assay | Thermo Fisher Scientific | Cat# 23,225 | |
| Commercial assay or kit | TNT Quick Coupled Transcription/ Translation System | Promega | Cat# L1170 | |
| Commercial assay or kit | NEXTflex ChIP-seq Library kit | Bioo Scientific | Cat# 5143–02 | |
| Commercial assay or kit | Magnetic mRNA isolation kit | NEB | Cat# S1550S | |
| Commercial assay or kit | NEXTflex Rapid directional RNA-seq kit | Bioo Scientific | Cat# 5138–08 | |
| Chemical compound, drug | Doxycycline | Sigma Aldrich | Cat# D9891 | |
| Chemical compound, drug | Hygromycin B | Gibco | Cat# 10687010 | |
| Chemical compound, drug | puromycin | Gibco | Cat# A11138-03 | |
| Chemical compound, drug | G418 | Gibco | Cat# 11811023 | |
| Chemical compound, drug | DharmaFECT 1 | Thermo Fisher Scientific | Cat# T-2001–03 | |
| Chemical compound, drug | FUGENE HD | Promega | Cat# E2311 | |
| Chemical compound, drug | Lipofectamine LTX with Plus Reagent | Invitrogen | Cat# 15338100 | |
| Chemical compound, drug | Turbofect | Thermo Fisher Scientific | Cat# R0533 | |
| Software, algorithm | Sequest Sorcerer platform | Sage-N Research | *Homo sapiens* protein sequence database (20,675 entries, UniProt) | |
| Software, algorithm | Scaffold | Proteome Software Inc. | Version 4.11.0 | |
| Software, algorithm | Protein Prophet algorithm | Protein Prophet algorithm | DOI: 10.1021/ac0341261 | |

