## [Editor Report]

This study describes a physical interaction between the Inositol polyphosphate multikinase enzyme (IPMK) and the SWI/SNF chromatin-remodeling complex. IPMK modulates SWI/SNF chromatin binding in mouse embryonic stem cells in particular at the transcription start sites of promoters with bivalent chromatin modifications. Loss of IPMK function led to defective activation of endodermal genes upon cell differentiation. This study identifies IPMK as a novel regulator of chromatin remodeling by SWI/SNF and gene expression in embryonic stem cells.

---

## [Decision Letter]

**Decision letter after peer review:**

Thank you for submitting your article "IPMK physically binds to the SWI/SNF complex and modulates BRG1 occupancy" for consideration by *eLife*. Your article has been reviewed by 3 peer reviewers, including Irwin Davidson as Reviewing Editor and Reviewer #1, and the evaluation has been overseen by Jessica Tyler as the Senior Editor.

As you can see from the comments below the referees raised several major issues that have to addressed in any revised version.

1). The referees were concerned that effects are seen only at sites with low BRG1 occupancy. To be sure that this is meaningful the referees propose in their comments below a series of further analyses of the data that should be performed.

2). SiRNA IPMK depletion is of limited efficiency with several blots showing an incomplete knockdown. This may lead to an under estimation of the possible impact IPMK has on BRG1 function. The authors should perform a full IPMK KO using CRISPR and re-assess the effects on BRG1 occupancy and gene expression. Alternatively, at the very least the authors should use a constitutive shRNA-mediated IPMK silencing to achieve a better reduction in its expression.

3). A major issue is the lack of a real functional readout on IPMK knockdown in terms of the biology of the ES cells. If as claimed genes regulated by bivalent promoters are the preferred targets this should translate into altered ES cell differentiation. The authors should investigate in detail the impact of IPMK silencing or knockout on ES cell biology to go beyond the analyses of gene expression.

Any revised version must also address the other issues raised by the referees.

*Reviewer #1 (Recommendations for the authors):*

Several issues should be addressed.

The authors describe the effect of IPMK depletion at transcriptional start sites of bivalent and H3K4me3 only promoters. Can the authors perform the same analyses at enhancers? According to their model, enhancers show higher BRG1 occupancy, and therefore should not be affected by IPMK depletion. However, this may not be true for all enhancers, there may be a subset with lower occupancy where IPMK depletion would have a comparable effect to that seen at the TSS. The authors must also take into account that the siRNA mediated depletion is of limited efficiency, several blots showing an incomplete knockdown. Perhaps if a full IPMK KO was made using CRISPR, even sites with stronger BRG1 occupancy would now be affected. Is this possible? See below.

As IPMK depletion affects sites with lower BRG1 occupancy, it is important to exclude the possibility that IPMK depletion results in reduced expression of BRG1 or another limiting subunit leading to reduced SWI/SNF levels.

The authors show that IPMK depletion leads to a more pronounced effect on expression of genes with bivalent promoters compared to H3K4me3 only promoters. As bivalent promoters are those that are destined to be activated or repressed upon ES differentiation, this would suggest that IPMK depletion should impact ES differentiation. The authors should test this idea and ask whether IPMK depletion by siRNA or, even better shRNA to achieve stable depletion, affects ES cell differentiation. Also, the authors make no comments on how IPMK siRNA and de-regulation of around 500 genes affects ES cell physiology or proliferation. This is relevant if a CRISPR KO was to be made, are the ES cells viable or not. If yes, a KO would be a useful complement as mentioned above to assess the full effect of IPMK on SWI/SNF function and subsequent ES differentiation. These data would strongly strengthen the paper and provide a functional readout for the how IPMK affects SWI/SNF beyond gene expression assays.

Can the authors address whether the IPMK effects on SWI/SNF genome occupancy and gene expression require its enzymatic activity or whether the effects are independent of enzymatic activity?

The manuscript requires thorough re-writing and proofing by a native English speaker as the text is often difficult to follow. The study could be written in a much clearer manner.

*Reviewer #2 (Recommendations for the authors):*

1. There is only one concern about the immunoprecipitations. The BAF155 co-IP in Figure 2C is not very convincing. There is a band in the IgG control that is stronger than the band in the IP lane.

2. Details on analysis of peaks for cut and run and ATAC-seq are missing. How were the changes in peak height quantified and determined to be statistically significant.

3. The rationale and method for defining clusters and then re-clustering should be clarified.

4. RNA-seq profiles with IPMK knockdown showed that some genes occupied by BRG1 and regulated at the chromatin level are also regulated by IPMK. However, there is no information about whether the effects of IPMK depletion on gene expression results from suppression of BRG1 occupancy. Experiments to address whether BRG1 depletion affects the expression of Phactr3, Nmra1l or any of the IPMK-regulated genes determined by RNA-seq should be conducted.

5. The manuscript loses focus by conducting RNA-seq in cells with Smarcb1 depletion. These data should be integrated with the data on BRG1 occupancy and chromatin accessibility. A key question to address is whether any of the overlapping Smarcb1/IPMK regulated genes can be associated with changes in BRG1 occupancy and chromatin accessibility upon IPMK and BRG1 knockdown.

*Reviewer #3 (Recommendations for the authors):*

To test the quality of the BRG1 CUT&RUN experiments, the authors could use the lists of typical enhancers and super-enhancers published in Whyte et al., 2013 (http://dx.doi.org/10.1016/j.cell.2013.03.035) and generate heat maps of BRG1 distribution showing separately each replicate of BRG1 CUT&RUN, as well as the non-specific IgG CUT&RUN control. For the comparison, they could use the previously validated GSM1581286 BRG1 ChIP-seq data.

To verify that their interpretation of low amplitude changes in ChIP-seq or CUT&RUN signal is correct, the authors should generate heat-maps as in Figure 4C and 4G showing individual replicates of both BRG1 CUT&RUN and ATAC-seq.

Similarly, the authors should also produce heat-maps of the five clusters of promoters shown in figure 5D, showing each BRG1 CUT&RUN replicate experiment, as well as non-specific IgG CUT&RUN control.

These figures will be determinant to test the validity of the differences in BRG1 enrichment observed in IPMK-depleted cells.

The description of the results of the ATAC-seq experiments at promoter elements is difficult to follow: multiple analysis are presented using different series of clusters of TSS. The authors should try to simplify this part of manuscript.

[Editors’ note: further revisions were suggested prior to acceptance, as described below.]

Thank you for resubmitting your work entitled "Inositol polyphosphate multikinase physically binds to the SWI/SNF complex and modulates BRG1 occupancy in mouse embryonic stem cells" for further consideration by *eLife*. Your revised article has been evaluated by Jessica Tyler (Senior Editor) and a Reviewing Editor.

The manuscript has been improved but there are some remaining issues that need to be addressed, as outlined below:

In this revised version, the authors have performed a number of important experiments to address previous concerns. These new data have strengthened the study and the conclusions. Specifically, they have used stable shRNA-mediated IPMK knockdown in addition to the siRNA and observed analogous effects on BRG1 occupancy. They have performed rescue experiments that show that the enzymatic activity of IPMK is not required for BRG1 occupancy and they have differentiated the shIPMK cells as embryoid bodies and provide evidence for downregulated expression of several endodermal markers. They have incorporated in their analyses BRG1 occupancy at previous lists of mESC enhancers. This new data addressed many of the major issues raised in the previous version.

Nevertheless, the authors should perform some additional revisions as several sections of the manuscript are rather difficult to follow.

1). As previously suggested, the authors should remove the data on siSMARCB1 in Figure 6 and Figure 6-supplement. This data interrupts the flow of the paper and dilutes the message. There are no analyses of how siSMARCB1 affects BRG1 or IPMK occupancy or chromatin accessibility by ATAC-seq and therefore these results cannot be properly interpreted. Also, there is no relevance in this paper for 3T3 cells. Removing this data does not affect the major conclusions of the paper and will make the paper easier to read.

2). The description of the data in Figure 5 and the Figure 5 supplements is extremely difficult to follow. All of this section needs careful attention and should be rewritten. For example, it is difficult to understand why total BRG1 loss at cluster 1 has a differential effect at bivalent and K4me3-only promoters. Why does it not impact the accessibility of the K4me3-only promoters. Figures 5F and G are particularly difficult to understand. Can this data be represented in another manner, or at least the description in the text should be improved?

3). The new experiments show reduced expression of endodermal markers in embryoid bodies generated from the shIPMK cells. The authors should comment on whether the genes that are not properly induced in the embryoid bodies displayed bivalent promoters in the ESCs and if yes, did they belong to a specific cluster as defined in Figure 5. In other words is there a relationship between altered BRG1 occupancy at the promoters in ESCs and failure to be properly induced in embryoid bodies.

4). In the discussion, the authors should avoid unnecessary speculation about how IPMK may affect SWI/SNF structure (lines 510-550). The discussion should be simplified and shortened to specifically address the results of this study (see lines 574-605).

5). Figure 4 supplement 2D should be brightened.

6). I would suggest that Figure 7B be transferred to the Figure supplement and that panels A and D of the Figure 7 supplement be transferred to the main Figure 7.

7). In general, the manuscript still requires a thorough revision by a native speaker as there are still many errors with many difficult to understand sentences.

---

## [Author Response]

1). The referees were concerned that effects are seen only at sites with low BRG1 occupancy. To be sure that this is meaningful the referees propose in their comments below a series of further analyses of the data that should be performed.

We agree with the fact that IPMK depletion decreased the BRG1 intensity majorly at the lowly-enriched BRG1 peaks but had no significant effect at the highly-enriched BRG1 peaks (Figure 4B). To prove that this effect is meaningful, we performed a series of further analyses suggested by the reviewers. Most importantly, we established two types of stable IPMK-depleted mESCs via shRNA-mediated *Ipmk* silencing (Figure 4C and Figure 4—figure supplement 2C, sh*Ipmk1* and sh*Ipmk2*) and performed BRG1 CUT&RUN assays. Consistent with the results from the siRNA-mediated Ipmk silencing (Figure 4B), shRNA-mediated Ipmk silencing significantly reduced the BRG1 intensity at the lowly-enriched BRG1 peaks but had no effect at the highly-enriched BRG1 peaks (Figure 4D). The reproducible results observed both in siRNA-mediated and shRNA-mediated silencing indicate that the decrease in the BRG1 intensity majorly at low BRG1 occupancy upon IPMK depletion is not an artifact. Instead, IPMK has a significant role in modulating BRG1 occupancy particularly for lowly-enriched BRG1 peaks.

2). SiRNA IPMK depletion is of limited efficiency with several blots showing an incomplete knockdown. This may lead to an under estimation of the possible impact IPMK has on BRG1 function. The authors should perform a full IPMK KO using CRISPR and re-assess the effects on BRG1 occupancy and gene expression. Alternatively, at the very least the authors should use a constitutive shRNA-mediated IPMK silencing to achieve a better reduction in its expression.

As mentioned above, we established two types of constitutive IPMK-depleted mESCs via shRNA-mediated *Ipmk* silencing (Figure 4C and Figure 4—figure supplement 2C, sh*Ipmk1* and sh*Ipmk2*). Compared with the efficiency of the siRNA-mediated *Ipmk* silencing (*Ipmk*KD), we observed that the shRNA-mediated knockdown efficiency was higher (compare Figure 4A-siRNA vs. Figure 4C-shRNA, and also compare Figure 2F and Figure 2—figure supplement 1D -siRNA vs. Figure 4—figure supplement 2C -shRNA). We re-assessed the IPMK depletion effects on BRG1 occupancy and gene expression by using the newly generated cell lines.

3). A major issue is the lack of a real functional readout on IPMK knockdown in terms of the biology of the ES cells. If as claimed genes regulated by bivalent promoters are the preferred targets this should translate into altered ES cell differentiation. The authors should investigate in detail the impact of IPMK silencing or knockout on ES cell biology to go beyond the analyses of gene expression.

To identify the function of IPMK depletion in terms of ESC biology, we performed embryoid body (EB, in vitro cell clumps recapitulating the early events of embryogenesis) formation and assessed the expression of differentiation markers (Figures 7A, B, and Figure 7—figure supplement 1A-C).

Any revised version must also address the other issues raised by the referees.Reviewer #1 (Recommendations for the authors):Several issues should be addressed.The authors describe the effect of IPMK depletion at transcriptional start sites of bivalent and H3K4me3 only promoters. Can the authors perform the same analyses at enhancers? According to their model, enhancers show higher BRG1 occupancy, and therefore should not be affected by IPMK depletion. However, this may not be true for all enhancers, there may be a subset with lower occupancy where IPMK depletion would have a comparable effect to that seen at the TSS. The authors must also take into account that the siRNA mediated depletion is of limited efficiency, several blots showing an incomplete knockdown. Perhaps if a full IPMK KO was made using CRISPR, even sites with stronger BRG1 occupancy would now be affected. Is this possible? See below.

To address this issue, we employed the previously determined mESC enhancer lists as reviewer #3 suggested (Whyte et al., 2013, http://dx.doi.org/10.1016/j.cell.2013.03.035) and aligned them with previously published BRG1 ChIP-seq peaks (GSE64825) and our BRG1 CUT&RUN peaks. As expected, highly-enriched BRG1 peaks (both ChIP-seq and CUT&RUN peaks) were primarily localized at enhancers (Figures 4B, D, and Figure 4—figure supplement 1E, upper part of the heatmap termed as “High”). Notably, IPMK depletion (both siRNA- and shRNA-mediated silencing) did not exhibit significant changes in BRG1 occupancy at these BRG1-enriched enhancers (Figures 4B, D, and Figure 4—figure supplement 1F).

As IPMK depletion affects sites with lower BRG1 occupancy, it is important to exclude the possibility that IPMK depletion results in reduced expression of BRG1 or another limiting subunit leading to reduced SWI/SNF levels.

The observed reduction in BRG1 intensity at the low BRG1 peaks (Figures 4B and D, bottom part of the heatmap termed as “Low”) were not derived from any changes in the expression of BRG1 or other subunits of the SWI/SNF complex (Figure 2—figure supplement 1D, Figure 4—figure supplement 1B, 2B and C), excluding the possibility that IPMK depletion results in reduced expression of BRG1 or another SWI/SNF complex subunit leading to the reduced BRG1 occupancy at the low BRG1 peaks.

The authors show that IPMK depletion leads to a more pronounced effect on expression of genes with bivalent promoters compared to H3K4me3 only promoters. As bivalent promoters are those that are destined to be activated or repressed upon ES differentiation, this would suggest that IPMK depletion should impact ES differentiation. The authors should test this idea and ask whether IPMK depletion by siRNA or, even better shRNA to achieve stable depletion, affects ES cell differentiation. Also, the authors make no comments on how IPMK siRNA and de-regulation of around 500 genes affects ES cell physiology or proliferation. This is relevant if a CRISPR KO was to be made, are the ES cells viable or not. If yes, a KO would be a useful complement as mentioned above to assess the full effect of IPMK on SWI/SNF function and subsequent ES differentiation. These data would strongly strengthen the paper and provide a functional readout for the how IPMK affects SWI/SNF beyond gene expression assays.

To address these comments, we employed shRNA-mediated *Ipmk* silencing.

1) We first confirmed that shRNA-mediated IPMK depletion significantly reduced the BRG1 intensity at bivalent promoters (Figure 5—figure supplement 1A and B), consistent with the results from the siRNA-mediated *Ipmk* silencing. These results support that IPMK depletion causes more pronounced effects on bivalent promoters (compared to H3K4me3-ONLY promoters).

2) To investigate whether IPMK depletion (which impacts the BRG1 occupancy and chromatin accessibility at bivalent promoters) impacts ES differentiation, we measured the RNA levels of key marker genes related to pluripotency and three germ layers (ectoderm, mesoderm, and endoderm) at various time points during embryoid body (EBs) formation. Notably, endodermal marker genes were dramatically downregulated in IPMK-depleted EBs (Figure 7A). By integrating the expression profiles of three types of marker genes, we found that IPMK depletion caused the most severe impact on the endodermal marker genes by reducing their expression levels (Figure 7B). Collectively, these data show that IPMK depletion impacts ES differentiation (especially affecting endodermal marker gene expression), thereby validating the above comments of reviewer #1.

3) Lastly, we observed no significant difference in morphology, proliferation rate, or viability between shNT and sh*Ipmk* mESCs.

Can the authors address whether the IPMK effects on SWI/SNF genome occupancy and gene expression require its enzymatic activity or whether the effects are independent of enzymatic activity?

To address whether the enzymatic activity of IPMK is required for regulating BRG1 occupancy, we performed rescue experiments by transfecting *Ipmk*KD mESCs with DNA constructs of wild-type IPMK (+WT) or catalytically dead IPMK (+SA). We first confirmed the stable expression of the constructs (Figure 4—figure supplement 2F) and then performed BRG1 CUT&RUN assays. Consistent with our previous results (Figures 4B and D), we observed significantly reduced BRG1 intensity at low BRG1 peaks upon *Ipmk*KD (Figure 4—figure supplement 2G and H). Although transfection of WT IPMK (+WT) did not fully restore the BRG1 intensity, we observed significantly increased BRG1 intensity in the *Ipmk*KD+WT mESCs compared with the intensity in the *Ipmk*KD mESCs (Figure 4—figure supplement 2G and H). Interestingly, transfection of catalytically dead IPMK (*Ipmk*KD+SA) also exhibited significantly increased BRG1 intensity compared with the intensity in the *Ipmk*KD mESCs and did not show a large difference in BRG1 intensity compared with that in the *Ipmk*KD+WT mESCs (Figure 4—figure supplement 2G and H). These results indicate that the enzymatic activity of IPMK is not required for regulating BRG1 occupancy in mESCs. Thus, the physical interaction between IPMK and BRG1 is presumably crucial for appropriate BRG1 occupancy (see also the Discussion).

The manuscript requires thorough re-writing and proofing by a native English speaker as the text is often difficult to follow. The study could be written in a much clearer manner.

Thank you for your suggestion, we will proceed accordingly. We hope the revised manuscript will meet the requirements.

Reviewer #2 (Recommendations for the authors):1. There is only one concern about the immunoprecipitations. The BAF155 co-IP in Figure 2C is not very convincing. There is a band in the IgG control that is stronger than the band in the IP lane.

To resolve this issue, we performed additional BAF155 co-IP experiments and reassessed the representative data in Figure 2C. As shown in the current, updated Figure 2C, the results have become even more convincing that IPMK binds to BAFF155 (with less background signal in the IP lane relative to the IgG control than that in the original analysis).

2. Details on analysis of peaks for cut and run and ATAC-seq are missing. How were the changes in peak height quantified and determined to be statistically significant.

For the CUT&RUN peak analysis, we used MACS2 (Zhang et al., 2008) (callpeak option, *P*-value < 0.005) to identify the peaks of proteins of interest (-t option) by using IgG as the background (-c option). For the ATAC-seq peak analysis, we used the same condition as in the CUT&RUN analysis but without the background (-c) option (since there is no factor to use as a background in ATAC-seq). The intensities at the peaks (or defined regions) were calculated using bwtool (Pohl and Beato, 2014) (matrix 1:1:1 option) with the relevant sequencing data (bigwig files) as the input, and the statistical significance of the genome-wide difference between the samples was assessed using the Wilcoxon rank sum test (see also Figures 4B, D, and G). We also discussed this part both in Materials and methods and figure legends.

3. The rationale and method for defining clusters and then re-clustering should be clarified.

IBRG1 has been reported to differentially regulate chromatin accessibility depending on its binding positions; BRG1 localized at -1 nucleosome in wide NFR (median length 808 bp) of H3K4me3-Only and bivalent promoters enhances the chromatin accessibility of NFR, whereas BRG1 localized at +1 nucleosome in narrow NFR (median length 28 bp) of H3K4me3-only promoters tends to inhibit the chromatin accessibility of NFR in mESCs (de Dieuleveult et al., 2016). Accordingly, we defined the upstream (-400 to -30 bp from the TSS) and downstream (-30 to +300 bp from the TSS) regions, which correspond to -1 and +1 nucleosome, respectively (Figure 5A), and clustered the TSSs into five clusters depending on the differential BRG1 intensity at the up/downstream regions upon IPMK depletion (Figures 5C–E). We removed the re-clustering (previous Figure 5— figure supplement 1) section to clarify our message.

4. RNA-seq profiles with IPMK knockdown showed that some genes occupied by BRG1 and regulated at the chromatin level are also regulated by IPMK. However, there is no information about whether the effects of IPMK depletion on gene expression results from suppression of BRG1 occupancy. Experiments to address whether BRG1 depletion affects the expression of Phactr3, Nmra1l or any of the IPMK-regulated genes determined by RNA-seq should be conducted.

To address this issue, we first designed two siRNAs targeting different regions of *Brg1* and confirmed the successful knockdown of *Brg1* expression (Figure 7—figure supplement 1D, left). Notably, we observed that IPMK or BRG1 depletion down-regulated *Phactr3* (bivalent promoters) and endodermal marker genes (*Foxa2*, *Gata6*, and *Sox17*) in mESCs (Figure 7—figure supplement 1D), suggesting that IPMK regulates the expression of these genes through BRG1. In support, we detected that ATAC-seq signals (chromatin accessibility) were decreased similarly upon *Ipmk*KD and *Brg1*KD at two promoters (Figures 5F and G). Since the reduced BRG1 occupancy upon *Ipmk*KD partially mimics the *Brg1*KD, the similar result upon *Ipmk*KD or *Brg1*KD further supports that IPMK plays a vital role in chromatin accessibility at promoters-TSSs by regulating the BRG1 occupancy.

5. The manuscript loses focus by conducting RNA-seq in cells with Smarcb1 depletion. These data should be integrated with the data on BRG1 occupancy and chromatin accessibility. A key question to address is whether any of the overlapping Smarcb1/IPMK regulated genes can be associated with changes in BRG1 occupancy and chromatin accessibility upon IPMK and BRG1 knockdown.

We agree with the reviewer. Due to the low number of overlapping Smarcb1/IPMK-regulated genes, it is challenging to present “significant” changes in BRG1 occupancy and chromatin accessibility upon IPMK or BRG1 knockdown in a genome-wide manner. Despite these difficulties, most of our representative genes (Figures 6A, B, E, and F) are indeed members of overlapping Smarcb1/IPMK down-regulated genes (Figure 6H-Clusters 5 and 6J) (also see the Author response image 1 and Author response table 1 that were extracted from the mRNA-seq analysis-cufflinks). These representative genes are shown to be 1) decreased in BRG1 occupancy at their TSS upon IPMK knockdown and 2) decreased in ATAC-seq intensity at their TSS upon IPMK or BRG1 knockdown in the same manner (Figures 6A, B, E, and F). Since the reduced BRG1 occupancy upon IPMK knockdown partially mimics BRG1 knockdown, the similar ATAC-seq reduction upon IPMK or BRG1 knockdown further supports that IPMK plays a vital role in regulating the expression of overlapping Smarcb1/IPMK down-regulated genes by maintaining BRG1-mediated chromatin accessibility at promoters-TSSs.

**Author response table 1. sa2table1:** Table shows the actual values and calculated significance corresponding to Author response image 1.

*Genes*	*Figure 6*	*Relative Expression*	*Log2 (vs EgfpKD)*	*P-value (vs EgfpKD)*				
		*EgfpKD*	*lpmkKD*	*Smarcb1KD*	*lpmkKD*	*Smarcb1KD*	*lpmkKD*	*Smarcb1KD*
*Rasl12*	*A*	*1*	*0.272476547*	*0.538887496*	*-1.8758*	*-0.891947*	*0.00145*	*0.08695*
*Phactr3*	*B and E*	*1*	*0.582057348*	*0.480741151*	*-0.780767*	*-1.05667*	*0.05265*	*0.0278*
*Lrrc61*	*F*	*1*	*0.490901076*	*0.604801519*	*-1.0265*	*-0.725463*	*0.00215*	*0.01485*
*Arhgap44*	*F*	*1*	*0.602265871*	*0.491195499*	*-0.731528*	*-1.02563*	*0.03035*	*0.04435*

**Author response image 1. sa2fig1:** 

Reviewer #3 (Recommendations for the authors):To test the quality of the BRG1 CUT&RUN experiments, the authors could use the lists of typical enhancers and super-enhancers published in Whyte et al., 2013 (http://dx.doi.org/10.1016/j.cell.2013.03.035) and generate heat maps of BRG1 distribution showing separately each replicate of BRG1 CUT&RUN, as well as the non-specific IgG CUT&RUN control. For the comparison, they could use the previously validated GSM1581286 BRG1 ChIP-seq data.

We agree with the reviewer and understand their concerns. The major reason that our BRG1 CUT&RUN peak annotation did not detect enhancers was that enhancers were not specified in the peak annotation program (instead, enhancers were just included as intergenic regions). Since the peak enrichment (Figure 4E) and significance (Figure 4F) both depend on the length of genomic annotation (and the intergenic sites are in total much longer than the promoter-TSSs), these may be the reason why intergenic sites (enhancers) showed the lowest and the most insignificance in BRG1 CUT&RUN peaks (Figure 4E and F).

To resolve this issue, we followed the comments of the reviewer as below:

Since the peak annotation analysis (Figures 4E and F) did not specify the enhancers, which show a strong BRG1 enrichment (de Dieuleveult et al., 2016; Laurette et al., 2015; Nakayama et al., 2017), we used the previously determined enhancer lists of mESCs (Whyte et al., 2013) and aligned with the BRG1 peaks. Consistent with the previous findings, highly enriched BRG1 ChIP-seq peaks from the previous study (Figure 4—figure supplement 1E; these data are equivalent to the GSE64825 or GSM1581286 BRG1 ChIP-seq data that reviewer #3 suggested) and highly enriched BRG1 CUT&RUN peaks from our study (Figures 4B, D, and Figure 4—figure supplement 1F) were mostly localized at enhancers (upper part of the heatmap). Furthermore, both BRG1 peaks (ChIP-seq peaks from the previous study and our CUT&RUN peaks) showed similar enrichment patterns in promoters and enhancers (Figure 4—figure supplement 1G), validating our CUT&RUN experiments. Notably, we observed that IPMK was highly enriched at enhancers (Figure 4B).

To verify that their interpretation of low amplitude changes in ChIP-seq or CUT&RUN signal is correct, the authors should generate heat-maps as in Figure 4C and 4G showing individual replicates of both BRG1 CUT&RUN and ATAC-seq.

Following the comment of the reviewer, we generated heatmaps as in Figure 4D and 4G showing individual replicates of both BRG1 CUT&RUN (Figure 4—figure supplement 2A) and ATAC-seq (Figure 4—figure supplement 2I).

Similarly, the authors should also produce heat-maps of the five clusters of promoters shown in figure 5D, showing each BRG1 CUT&RUN replicate experiment, as well as non-specific IgG CUT&RUN control.These figures will be determinant to test the validity of the differences in BRG1 enrichment observed in IPMK-depleted cells.

Following the comment of the reviewer, we generated BRG1 CUT&RUN heatmaps of the five TSS clusters in H3K4me3-ONLY and bivalent promoters (Figures 5D and E). Due to the limited space, we could not add the BRG1 CUT&RUN replicates and non-specific IgG CUT&RUN controls in main figures (instead, BRG1 CUT&RUN replicates are shown in Figure 4—figure supplement 2A and IgG CUT&RUN controls are shown in average line plots in Figures 5D and E).

The description of the results of the ATAC-seq experiments at promoter elements is difficult to follow: multiple analysis are presented using different series of clusters of TSS. The authors should try to simplify this part of manuscript.

We agree with the reviewer. We have removed the additional analysis (previous Figure 5—figure supplement 1) section to clarify our message.

[Editors’ note: further revisions were suggested prior to acceptance, as described below.]

1). As previously suggested, the authors should remove the data on siSMARCB1 in Figure 6 and Figure 6-supplement. This data interrupts the flow of the paper and dilutes the message. There are no analyses of how siSMARCB1 affects BRG1 or IPMK occupancy or chromatin accessibility by ATAC-seq and therefore these results cannot be properly interpreted. Also, there is no relevance in this paper for 3T3 cells. Removing this data does not affect the major conclusions of the paper and will make the paper easier to read.

As suggested, we removed the data on si*Smarcb1* and NIH3T3 cells in Figure 6G-J and Figure 6—figure supplement 1B-F. We also removed the corresponding parts in the manuscript (lines 417-462 in the original manuscript).

2). The description of the data in Figure 5 and the Figure 5 supplements is extremely difficult to follow. All of this section needs careful attention and should be rewritten. For example, it is difficult to understand why total BRG1 loss at cluster 1 has a differential effect at bivalent and K4me3-only promoters. Why does it not impact the accessibility of the K4me3-only promoters. Figures 5F and G are particularly difficult to understand. Can this data be represented in another manner, or at least the description in the text should be improved?

We improved the description in our revised manuscript. Regarding the reviewer’s question (although IPMK depletion reduced BRG1 occupancy at both promoters, why does it impact the chromatin accessibility of the bivalent promoters more than the H3K4me3-Only promoters?), there are three possible answers.

A) Consistent with our results, de Dieuleveult et al., (2016, http://dx.doi.org/10.1038/nature16505) showed that Brg1 depletion reduces the ATAC-seq (chromatin accessibility) signals at bivalent promoters in mouse embryonic stem cells (mESCs). The line plots are Extended Data Figure 8 from the previous study (de Dieuleveult et al., 2016). Comparing the ATAC-seq signals between H3K4me3-Only (top-left) and bivalent (bottom-left) promoters, Brg1 depletion (which would cause total BRG1 loss at both promoters, just like Cluster1 in our study) impacts the ATAC-seq signals more at bivalent promoters than H3K4me3-Only promoters.

B) When comparing the BRG1 occupancy in *Ipmk*KD cells (Figure 5D and E, red lines) at Cluster1 in two promoter types, we observed that H3K4me3-Only promoters contain more BRG1 occupancy than bivalent promoters. In other words, although Cluster1 exhibits total BRG1 loss in *Ipmk*KD cells, there remains some degree of BRG1 occupancy at H3K4me3-Only promoters, which exhibit higher levels than the bivalent promoters (Figure 5D and E). For direct comparisons, we drew line plots comparing the BRG1 occupancy in *Ipmk*KD cells at two promoter types and found that H3K4me3-Only promoters contain higher BRG1 occupancy than bivalent promoters at all clusters (Figure 5—figure supplement 1C). Thus, the remaining BRG1 at H3K4me3-Only promoters may maintain the chromatin accessibility, causing the differential effect between the two promoter types.

C) Lastly, de Dieuleveult et al., (2016, http://dx.doi.org/10.1038/nature16505) previously showed that the majority of chromatin remodelers (BRG1, CHD1, CHD2, CHD4, CHD6, CHD8, CHD9, and EP400) exhibit higher enrichment at H3K4me3-Only promoters than bivalent promoters (Extended Data Figure 3 and 5 in de Dieuleveult et al., 2016). To test this in our study, we drew line plots comparing the chromatin remodeler occupancy at two promoter types in five clusters using the data sets from the previous study (GSE64825, de Dieuleveult et al., 2016). Consistent with the previous findings, all of the chromatin remodelers (CHD1, CHD2, CHD4, CHD6, CHD8, CHD9, and EP400) exhibited higher occupancy at H3K4me3-Only promoters than bivalent promoters (Figure 5—figure supplement 1D). Thus, at H3K4me3-Only promoters, these highly enriched chromatin remodelers may compensate for the loss of BRG1 upon IPMK depletion, thereby maintaining the chromatin accessibility.

Taken together, these three possibilities support the differential chromatin accessibility at two promoter types in terms of *Ipmk*KD-driven reduced BRG1 occupancy. We added the figures and descriptions to our revised manuscript (lines 309-400).

3). The new experiments show reduced expression of endodermal markers in embryoid bodies generated from the shIPMK cells. The authors should comment on whether the genes that are not properly induced in the embryoid bodies displayed bivalent promoters in the ESCs and if yes, did they belong to a specific cluster as defined in Figure 5. In other words is there a relationship between altered BRG1 occupancy at the promoters in ESCs and failure to be properly induced in embryoid bodies.

All of the reduced endodermal markers (*Foxa1*, *Foxa2*, *Gata6*, and *Sox17*) in sh*Ipmk* embryoid bodies (EBs) displayed bivalent promoters in the mESCs. *Foxa2*, *Gata6*, and *Sox17*, but not *Foxa1*, exhibited decreased BRG1 occupancy at their promoters upon IPMK depletion; *Foxa2* and *Gata6* were included in Cluster2, whereas *Sox17* was included in Cluster3 (Figure 5E). *Foxa1* did not exhibit changes in BRG1 occupancy upon IPMK depletion. Taken together, these results indicate that the *Ipmk*KD-driven reduction in BRG1 occupancy at the bivalent promoters of endodermal markers in mESCs may result in their failure to be appropriately induced in EBs. We added this comment in our revised manuscript (lines 445-457).

4). In the discussion, the authors should avoid unnecessary speculation about how IPMK may affect SWI/SNF structure (lines 510-550). The discussion should be simplified and shortened to specifically address the results of this study (see lines 574-605).

We removed the speculation (lines 510-550 and 574-605) from the discussion and the corresponding data (model on the right in Figure 7C and Figure 7—figure supplement 1E). Overall, we simplified and shortened the discussion in the revised manuscript (lines 459-511).

5). Figure 4 supplement 2D should be brightened.

As suggested, we brightened Figure 4—figure supplement 2D by changing the Log_2_(Ipmk depletion/Control) threshold from (-1, +1) to (-0.8, +0.8).

6). I would suggest that Figure 7B be transferred to the Figure supplement and that panels A and D of the Figure 7 supplement be transferred to the main Figure 7.

As suggested, we moved Figure 7B to Figure 7—figure supplement 1C and moved Figure 7—figure supplement 1A and D to Figure 7A and C, respectively.

7). In general, the manuscript still requires a thorough revision by a native speaker as there are still many errors with many difficult to understand sentences.

As suggested by the reviewer, the manuscript was revised by a native English speaker.